# Understanding the effect of fire on vegetation composition and gross primary production in a semi-arid shrubland ecosystem using the Ecosystem Demography (EDv2.2) model

Karun Pandit[1], Hamid Dashti[2], Andrew T. Hudak[3], Nancy F. Glenn[4], Alejandro N. Flores[4], and Douglas J. Shinneman[5]

[1]School of Forest Resources and Conservation, University of Florida, 1745 McCarty Drive, Gainesville, FL 32611, USA
[2]School of Natural Resources and the Environment, University of Arizona, 1064 East Lowell Street, Tucson, AZ 8572, USA
[3]U.S. Forest Service, Rocky Mountain Research Station, 1221 South Main Street, Moscow, ID 83843, USA
[4]Department of Geosciences, Boise State University, 1910 University Dr, Boise, ID 83725, USA
[5]U.S. Geological Survey, Forest and Rangeland Ecosystem Science Center, 970 Lusk St., Boise, ID 83706, USA

**Correspondence:** Karun Pandit (karunpandit@gmail.com)

**Abstract.** Wildfires in sagebrush (*Artemisia* spp.) dominated semi-arid ecosystems in the western United States have increased dramatically in frequency and severity in the last few decades. Severe wildfires often lead to the loss of native sagebrush communities and change the biogeochemical conditions which make it difficult for sagebrush to regenerate. Invasion of cheat-grass (*Bromus tectorum*) accentuates the problem by making the ecosystem more susceptible to frequent burns. Managers
have implemented several techniques to cope with the cheatgrass-fire cycle, ranging from controlling undesirable fire effects by removing fuel loads either mechanically or via prescribed burns, to seeding the fire-affected areas with shrubs and native perennial forbs. There have been a number of studies at local scales to understand the direct impacts of wildfire on vegetation, however there is a larger gap in understanding these impacts at broad spatial and temporal scales. This need highlights the importance of dynamic global vegetation models (DGVMs) and remote sensing. In this study, we explored the influence of fire
on vegetation composition and gross primary production (GPP) in the sagebrush ecosystem using the Ecosystem Demography (EDv2.2) model, a dynamic global vegetation model. We selected Reynolds Creek Experimental Watershed (RCEW) to run our simulation study, an intensively monitored sagebrush-dominated ecosystem in the northern Great Basin. We ran point-based simulations at four existing flux-tower sites in the study area for a total 150 years after turning on the fire module in the 25[th] year. Results suggest dominance of shrubs in a non-fire scenario, however under the fire scenario we observed contrasting
phases of high and low shrub density and $C_3$ grass growth. Regional model simulations showed a gradual decline in GPP for fire-introduced areas through the initial couple of years instead of killing all the vegetation in the affected area in the first year itself. We also compared the results from EDv2.2 with satellite-derived GPP estimates for the areas in RCEW burned by a wildfire in 2015 (Soda Fire). We observed moderate pixel-level correlations between maps of post-fire recovery EDv2.2 GPP and MODIS derived GPP. This study contributes to understanding the application of ecosystem models to investigate temporal
dynamics of vegetation under alternative fire regimes and post-fire ecosystem restoration.

# 1 Introduction

The number and intensity of wildfires in the sagebrush-steppe of the semi-arid Great Basin, western US, have increased dramatically (Keane et al., 2008). Studies have shown that sagebrush (*Artemisia* spp.) has declined significantly across the Great Basin due to fire and other disturbances (Knick et al., 2003; Pilliod et al., 2017; Rigge et al., 2019; Schroeder et al., 2004). The low stature of sagebrush makes it less adapted in morphological terms to survive fires as most of the flammable fuels are close to the ground (Hood and Miller, 2007; McArthur and Stevens, 2004; Welch and Criddle, 2003). In addition, ongoing research indicates that sagebrush regeneration is complicated by changes in climate, long germination and growth times, and seed dispersal (Chambers, 2000; Shriver et al., 2018; Walton et al., 1986). Even though fire is often recognized as a natural ecosystem process, it reduces woody shrub biomass while increasing herbaceous biomass (Ellsworth et al., 2016). Invasion of non-native cheatgrass (*Bromus tectorum*) alters the competitive balance between woody and herbaceous plants, and also makes the ecosystem more susceptible to frequent and larger fires (Baker, 2006; Building et al., 2013; Whisenant, 1990). A recent study has shown that this cheatgrass-fire cycle has resulted in more than one-third of the Great Basin having been invaded by cheatgrass (Bradley et al., 2018), which represents an enormous community shift with potentially large yet unknown effects on ecosystem function at a regional scale (Bradley et al., 2006; Bradley, 2010; Fusco et al., 2019).

Land managers and scientists have identified potential techniques to cope with the problems related to the altered fire regime in the Great Basin, ranging from controlling fire incidents with removing fuel loads either mechanically or using prescribed burns, to seeding the burned areas with shrubs and native perennial forbs. There have been a number of studies (e.g., Diamond et al., 2012; Ellsworth et al., 2016; Miller et al., 2013; Murphy et al., 2013) at the local scale to understand fire impacts, with many studies suggesting fire suppression as a technique to preserve the sagebrush ecosystem. However, there is a gap in understanding the influence at broader spatial scales. Remote sensing studies provide contemporary insights of ecosystem changes at broad spatial scales (e.g., Bradley et al., 2018). However, longer temporal-scale studies in the context of future climate scenarios are needed to better understand fire effects on shrub dominated ecosystems like the sagebrush-steppe (Knutson et al., 2014; Nelson et al., 2014).

One method to consider long time scales in the effects of fire on sagebrush ecosystems is to utilize dynamic global vegetation models (DGVMs) (Lenihan et al., 2007; Li et al., 2012). A DGVM can be placed anywhere along the continuum of individual-based to area-based models (Fisher et al., 2010; Smith et al., 2001). Individual-based models (IBMs) represent vegetation at the individual plant level incorporating complex community processes like growth, mortality, recruitment, and disturbances. Area-based models, on the other hand, represent plant communities with area-averaged representation making them more efficient for broad scale applications (Bond-Lamberty et al., 2015; Fisher et al., 2010; Smith et al., 2001). DGVMs are now increasingly intertwined with land surface models in ways that facilitate the integrated simulation of changes in vegetation community composition and surface water, energy, and biogeochemical cycles in response to changes in climate, land use, and fire regimes. Fisher and Koven (2020) provide a review of the increasingly sophisticated treatment of land surface processes in global land models, highlighting in particular the complex ways that vegetation influences fluxes and stores of water, energy, and carbon within these models. In the last two decades, fire sub-models in various DGVMs have evolved through time

from simple statistical methods to more complicated approaches with induced ignition and process-based spread and intensity (Thonicke et al., 2001, 2010; Knorr et al., 2016).

Ecosystem Demography (EDv2.2) is a DGVM originally developed in 2001 (Moorcroft et al., 2001). EDv2.2 is a cohort-based model that seeks to balance the fidelity of process representation in individual-based models with the computational efficiency of area-based models, wherein individual plants with similar properties, in terms of size, age, and function, are

grouped together to reduce the computational cost while retaining most of the dynamics of IBMs (Fisher et al., 2010). Because of this balance between process fidelity and computational burden, demography-based models are becoming increasingly popular versions of DGVMs within global land models (Fisher et al., 2018). While EDv2.2 was originally developed for a tropical forest ecosystem, it has since been updated for broader use (Medvigy et al., 2009), including to understand fire behavior under different probable scenarios in tree dominated ecosystems (Trugman et al., 2016; Zhang et al., 2015).

In this study, we used the Ecosystem Demography model (EDv2.2) with a recently developed plant functional type (PFT) parameterization of shrubs (Pandit et al., 2019) with the objective to examine model-derived effects of fire on a shrubland ecosystem in the Reynolds Creek Experimental Watershed (RCEW), Idaho, USA. We developed and ran a two-step numerical experiment to accomplish this. First, we explored the projected gross primary production (GPP) of a sagebrush-steppe ecosystem (in terms of shrub and $C_3$ grass PFTs) in EDv2.2 for two different fire disturbance scenarios and a no-fire or control

scenario (performed at point-level). Second, we compared the model-simulated spatiotemporal variability of GPP to a remotely sensed estimate of GPP (Wylie et al., 2003; Running et al., 2004) prior to and after a 2015 fire that burned a portion of the RCEW study area.

## 2  Methods

### 2.1  Ecosystem Demography (EDv2.2) model

EDv2.2 is a process-based dynamic global vegetation model which takes cohorts (a group of individuals with similar properties) as the smallest units of simulation. It is composed of a series of gridded cells, which experience meteorological forcing from corresponding gridded data or from a coupled atmospheric model (Medvigy, 2006). It captures both vertical and horizontal distributions of vegetation structure and compositional heterogeneity better than most of the area-based models (Kim et al., 2012; Moorcroft et al., 2001; Moorcroft, 2003; Sellers et al., 1992). EDv2.2 has a fire subroutine which evaluates conditions

leading to potential fire ignition and quantifies fire disturbance effects on vegetation. A detailed description of the EDv2.2 model structure including its fire subroutine is available in earlier publications (Longo et al., 2019b; Moorcroft et al., 2001; Medvigy et al., 2009). Here we present a brief summary of the fire subroutine.

In this model, fire ignition probability is based on soil dryness which is local (within-gap) in origin but can spread into adjacent areas given favorable conditions for fire. Burn rate or fire severity is proportional to local fuel availability or total

aboveground biomass (AGB). Under the current model settings, all plants in a burnt patch are killed while parts of carbon and nitrogen are transferred into the below-ground biogeochemical module (Moorcroft et al., 2001). The area of burnt patches within grids can increase linearly through years as a function of aboveground biomass (AGB). New burnt patches are created

every year when the minimum area necessary to generate a new patch is available through the loss of affected cohorts. Along with other disturbance factors in EDv2.2, the fire sub-module creates and maintains age- and size-based heterogeneity at sub-grid levels to closely resemble a broad range of structure and composition in a disturbed ecosystem. For example, a study from South America by Longo et al. (2019a) showed that this model represented a fire-disturbed ecosystem like woody savanna very well. Users can adjust the dryness threshold for fire ignition and fire severity parameters (defined between 0 to 1) to determine the level of fire-related disturbance depending upon available fuel. The fire related disturbance rate ($\lambda_{\mu,\mu_0}^{FR}$) affecting patch $u$ (and potentially creating new patches $u_0$) is given by the following equation (Eq. 1) as originally defined by Moorcroft et al. (2001) and later revisited by Longo et al. (2019b).

$$\lambda_{\mu,\mu_0}^{FR} = I \sum_{u=1}^{N_p} \sum_{k=1}^{N_{T_u}} \left\{ \left[ C_{ul_k} + F_{AG_{uk}}(C_{u\sigma_k} + C_{uh_k}) \right] \gamma_u \alpha_u \right\} \tag{1}$$

where patches are denoted by subscript $u$, $N_p$ is number of patch, $N_{T_u}$ is number of cohort in patch where patches are denoted by $u$, $\gamma_u$ is the binary ignition function as defined in equation 2, $\alpha_u$ is relative area of patch $u$, $I$ is fire intensity, $F_{AG_{uk}}$ is fraction of tissue aboveground, $C_{ul_k}$ is leaf biomass, $C_{u\sigma_k}$ is sapwood biomass and $C_{uh_k}$ is structural biomass. The binary ignition function (Eq. 2) represents the local dryness of environment which depends on the average soil moisture within a chosen soil depth.

$$\gamma_u = \begin{cases} 1, & \text{if } \left( \frac{1}{|Z_{Fr}|} \int_{Z_{Fr}}^0 \nu_g dz \right) < \nu_{Fr} \\ 0, & \text{otherwise} \end{cases} \tag{2}$$

where, $\nu_g dz$ is soil moisture at given soil layer thickness $dz$, $Z_{Fr}$ is the maximum soil depth considered in analyzing dryness and $\nu_{Fr}$ is an average soil moisture below which ignition is assumed to occur.

## 2.2 Study area

We ran the EDv2.2 model at the Reynolds Creek Experimental Watershed (RCEW), located in the Northern Great Basin region of the western United States (Fig. 1a). RCEW is operated by the USDA Agricultural Research Service and is also a Critical Zone Observatory (CZO). The watershed is approximately 240 km$^2$ in area with elevation ranging from about 900 to 2200 m. With an increase in elevation, there is an increase in mean annual precipitation and a decrease in mean annual temperature (Flerchinger et al., 2020; Renwick et al., 2019). Mean annual temperature ranges from 5 to 10 °C and mean annual precipitation range from 250 to 1100 mm in the watershed. Because of the strong orographic gradient in temperature in the watershed, most precipitation at lower elevations falls as rain, whereas precipitation at higher elevations is dominated by snow. The higher elevations in the southern areas of the watershed are dominated by quaking aspen (*Populus termuloides*), Douglas fir (*Pseudotsuga menziesii*), and western juniper (*Juniperus occidentalis*) (Seyfried et al., 2000). The lower elevations are primarily covered with Wyoming big sagebrush (*Artemisia tridentata* ssp. *wyomingensis*), low sagebrush (*Artemisia arbuscula*), rabbitbrush (*Ericameria nauseosa*) and bitterbrush (*Purshia tridentata*). Perennial herbs like bluebunch wheatgrass (*Pseudoroegneria spicata*),

**Table 1.** Description of EC sites used in the point-based analysis.

| Site | Ameriflux ID | Location | Elevation [m] | Mean annual precipitation [mm] | Mean annual temperature [°C] |
|------|-------------|----------|---------------|-------------------------------|------------------------------|
| WBS | US-Rws | 43.1675, -116.7132 | 1425 | 290 | 8.9 |
| LS | US-Rls | 43.1439, -116.7356 | 1608 | 333 | 8.4 |
| US | US-Rwf | 43.1207, -116.7231 | 1878 | 505 | 6.5 |
| RMS | US-Rms | 43.0645, -116.7486 | 2111 | 800 | 5.4 |

needle and thread (*Hesperostipa comata*), western wheatgrass (*Pascopyrum smithii*), tapertip hawksbeard (*Crepis acuminata*), and yarrow (*Achillea millefolium*) are also present (Pyke et al., 2015). The 2015 Soda Fire burned over 1,000 $km^2$ in southeast Oregon and southwest Idaho, including approximately 32% of RCEW in its northern region (Fig. 1b). Collaborative efforts be-
tween federal, state and private agencies have been applied to assess risk and devise a plan to implement treatments to stabilize burned areas, promote recovery of native plant communities, increase perennial grasses, and reduce invasive annual species (BLM, 2016).

We used EDv2.2 to run both point-based and regional analyses in RCEW. For the point-based runs, we used four 200 m x 200 m polygons centered at four eddy covariance (EC) tower sites in RCEW to represent the tower footprints. The four sites
include: Wyoming Big Sagebrush (WBS), Lower sheep (LS), Upper Sheep (US), and Reynolds Mountain Sagebrush (RMS) (Table 1). Wyoming big sagebrush is the dominant shrub at the WBS site with perennial grasses like bluebunch wheatgrass (*Pseudoroegneria spicata*), squirreltail (*Elymus elymoides*), and Sandberg bluegrass (*Poa secunda*). The dominant shrub at the LS site is low sagebrush (*Artemisia arbuscula*) along with Sandberg bluegrass, squirreltail (*Elymus elymoides*), and Idaho fescue (*Fescue idahoensis*). Mountain big sagebrush (*Artemisia tridentata* ssp. *vaseyana*) is the common shrub cover at the US and
RMS sites, where there is also a strong presence of forbs including longleaf phlox (*Phlox longifolia*), pale agoseris (*Agoseris glauca*), and silvery lupine (*Lupinus argenteus*) (Flerchinger et al., 2020). For regional runs, we discretized the watershed into a 1 km rectangular grid covering the entirety of the watershed, consistent with the resolution of the meteorological forcings input to the model described below. The study area in the regional runs consisted of the Soda Fire region of RCEW (Soda Fire region contained within the black polygon in Fig. 1b) and the whole of the RCEW (contained within the black polygon in Fig.
1b).

## 2.3 Meteorological forcing data

Meteorological forcing data input to the EDv2.2 model consisted of output from a multi-decadal run of the Weather Research and Forecast (WRF) model (Skamarock et al., 2008), which was used to dynamically downscale data from the North American Regional Reanalysis (National Centers for Environmental Prediction, National Weather Service, NOAA, U.S. Department of
Commerce, 2005) to a spatial resolution of 1 km (Pandit et al., 2019) (Table 2). WRF outputs correspond to atmospheric outputs at a standard height of 2 m for temperature and specific humidity, 10 m for wind speed and direction, and the ground

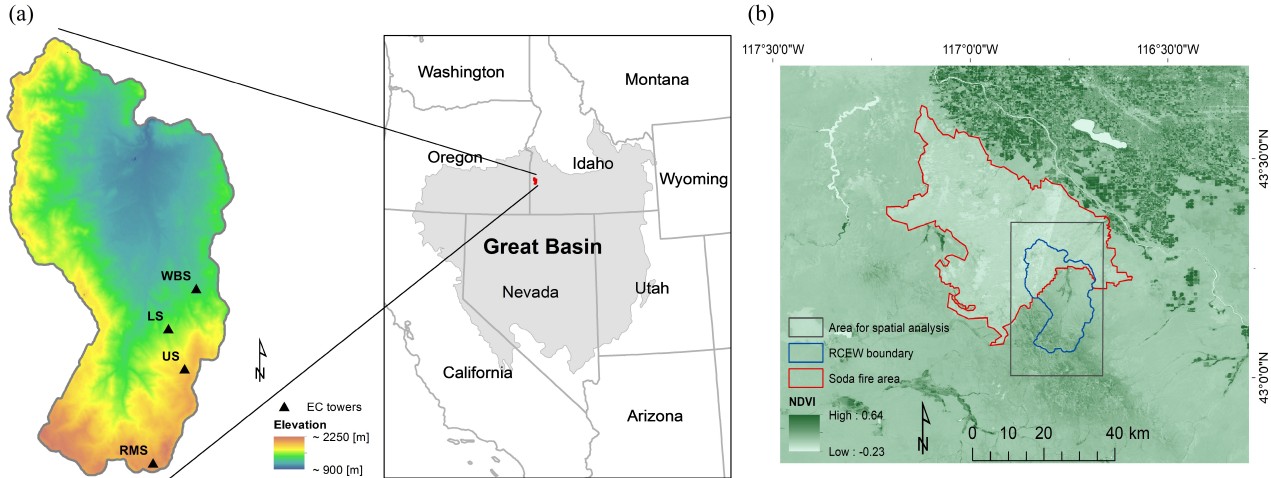

**Figure 1.** (a) Location of the four EC flux tower sites within the Reynolds Creek Experimental Watershed (RCEW) study area. The inset map shows the location of RCEW within the Northern Great Basin (LCC, 2019). The Great Basin area is shown in grey shading. (b) Map showing area affected by Soda Fire, 2015 (red polygon), boundary of RCEW (blue boundary), and rectangle covering RCEW (black polygon) used to run the regional EDv2.2 simulations. Normalized Difference Vegetation Index, NDVI (Landsat image, August, 2015) map in the background shows the disturbance from fire in the Soda Fire area.

surface for downward shortwave and longwave radiation, surface pressure, and precipitation (Flores et al., 2016). The temporal resolution of the WRF data is 1 hr and it is available for the period from October 1, 1986 to September 30, 2018. We partitioned shortwave radiation into direct and diffuse, visible and near-infrared components as summarized by Weiss and Norman (1985).

**Table 2.** Meteorological data from the WRF model used for simulation. Adapted from Pandit et al., (2019)

| Variable description | Name | Unit |
|---|---|---|
| 2-m temperature | T2 | K |
| Surface pressure | PSFC | Pa |
| Accumulated precipitation | RAINNC | mm |
| Terrain height | HGT | m |
| 10-m u wind (zonal) component | U10 | $\mathrm{ms^{-1}}$ |
| 10-m v wind (meridional) component | V10 | $\mathrm{ms^{-1}}$ |
| 2-m specific humidity | Q2 | $\mathrm{kgkg^{-1}}$ |
| Downward longwave flux at ground surface | GLW | $\mathrm{Wm^{-2}}$ |
| Downward shortwave flux at ground surface | SWDOWN | $\mathrm{Wm^{-2}}$ |

## 2.4 Multi-decadal simulation at point scale

We ran point-based simulations at four EC tower sites in RCEW to understand the multi-decadal temporal dynamics of PFTs for alternative fire conditions. We initialized ecosystem conditions using representative existing vegetation conditions with equal densities (0.25 plants $m^{-2}$) of shrubs and grasses as PFTs. The shrub density was based on field studies in the area (Glenn et al., 2017). For the shrubs, we used a PFT especially developed for sagebrush in the study area based on our previous work (Pandit et al., 2019) whereas for the grasses, we used the temperate $C_3$ grass PFT which is the closest match from among available PFTs in EDv2.2. We assumed that this existing temperate grass PFT in the model would represent common perennial grass species in the study area. We minimized interannual climate variability by calculating mean monthly precipitation from thirty years of WRF data (1988-2017), then selecting the year 2012 as the year that most closely matched the 30-year mean precipitation record. All four sites were run for an initial 25 years, after which each site was run with three different scenarios: (i) no fire, (ii) low fire severity, and (iii) high fire severity, for the next 125 years. In the fire scenario simulations, we ran the model with active fire for these later 125 years. The fire severity parameter in the model which specifies intensity of disturbance from fire can range from 0 to 1, where we applied 0.5 and 0.9 values for low and high severity fires, respectively. We observed GPP trends of shrub and grass PFTs for these three scenarios at all four EC sites, and compared results with GPP data from the sites (Fellows et al., 2017).

## 2.5 Multi-year simulation at regional scale

We performed regional (watershed) scale simulations to perform comparisons across simulations for fire/no-fire conditions, and between model simulations and satellite-derived estimates of ecosystem productivity. First, we compared the fire caused vegetation disturbance and recovery at the regional scale by allowing EDv2.2 to run with both fire and no-fire (control) conditions. Second, we compared the model predicted GPP (for both burnt and unburnt areas in the region) with MODIS derived GPP from the study area. To perform these simulations, we initialized EDv2.2 with a near-bare-earth scenario of 0.1 plants $m^{-2}$ for all allowed PFTs (i.e. $C_3$ grass, shrub, northern pines and late conifers) from 1990 and ran it for the following 25 years. Our analysis indicated that 25 years of spin-up was sufficient for GPP to reach equilibrium (Fig. S1 in the Supplement). For these model runs, we used meteorological data from the years corresponding with the simulation years, except for 2018 and 2019 when WRF data were not available. For these two years, we imputed WRF data from other years which closely resembled monthly total precipitation with the observations (NOAA, 2019).

For the first experiment, we ran fire and no-fire model simulations for a region inside RCEW which was affected by the Soda Fire in 2015 (hereafter Soda Fire scenario). For the fire scenario, we activated the fire subroutine in the model from 2015 and ran it until 2019. In this run, we adopted a high fire severity (0.9) to relate closely with the severity observed in the Soda Fire. For the no-fire (control) scenario, we allowed the model to continue without fire until 2019. We compared differences between the fire and no-fire simulations for each year.

For the next experiment, we ran EDv2.2 in a manner that would best represent the true circumstances for the entire study area (hereafter RCEW scenario). To perform this, we introduced fire (with same parameter as above) only into that portion of

RCEW which actually burned in 2015 and simulated the remaining portion of the watershed without fire. The purpose of this experiment was to compare the predictions from EDv2.2 (for burnt and unburnt areas) with that derived from MODIS images. The unburnt area in this simulation is used as a benchmark for comparisons and to offset annual variations. Like before, we ran the model with these conditions for the next 5 years (2015 to 2019). We produced GPP from MODIS GPP CONUS datasets (Robinson et al., 2018), using Google Earth Engine. The mean of all available MODIS images for July of each year was calculated, clipped and resampled to match the spatial coverage and grid resolution (1 km) of the EDv2.2 simulation, before comparing them against simulated mean monthly GPP values of July from the model.

## 3   Results

### 3.1   Multi-decadal GPP prediction at point scale

Temporal dynamics of the GPPs for shrub and $C_3$ grass PFTs were similar for the LS, WBS and US sites while slightly different for the RMS site (Fig. 2), which is located at higher elevation (Fig. 1a). Without fire, shrubs eventually dominated to comprise the entirety of GPP persisting through the end of the simulation period. GPP for $C_3$ grass was high during the initial years, but decreased rapidly after about 2-3 years of simulation, while shrub GPP increased gradually and became more dominant than grass after 10-15 years. Between 30 and 40 years, shrub GPP peaked, $C_3$ grass GPP completely disappeared, and GPP reached an approximate equilibrium at or slightly above 0.3 $\mathrm{kgCm^{-2}yr^{-1}}$ for the three lower elevation sites (LS, US, WBS) and at about 0.55 $\mathrm{kgCm^{-2}yr^{-1}}$ for the highest elevation site (RMS). We observed that during its initial rapid growth phase (Fig. 2), some portion of the total above ground biomass (AGB) is also covered by $C_3$ grass (Fig. A1), which in the latter years was completely wiped out by shrub AGB. We did not observe any growth of conifer PFTs throughout the simulation period, even for the no fire scenario.

Upon activation of fire module after 25 years of simulation, shrub GPP declined abruptly and $C_3$ grass GPP increased dramatically in all four study sites. However, around 25 years after fire activation, shrubs initiate a recovery and maintain a gradual increase until reaching a peak in 50-75 years; at the same time $C_3$ grass GPP gradually decreased to a minimal level. We observed lower overall GPP during the years when shrub GPP was at the peak, since at this time $C_3$ grass productivity was at the minimum. Disturbance rates from fire spiked in the first couple of years when fire was first introduced and later stabilized to closely follow the trend of shrub AGB (Fig. A1), suggesting the highest disturbance rate at the peak of shrub AGB leading to decline of shrub GPP (and shrub AGB) afterwards. A similar cycle was observed for the remainder of the simulation years. In most of the cases, we observed the peaks of total GPP approaching total GPP from the no fire scenario (at a cycle of about 60-75 years). For most of the sites, while shrub GPP remained lower compared to the no fire scenario $C_3$ in the post-fire years, grass GPP dominates the overall shape of total GPP. However, cycles of total AGB after fire matched well with the trend of shrub AGB which in turn influence the approximate fire return interval (with maximum fire disturbance rate in about 50-75 years) in the ecosystem.

We identified some differences between low and high fire severity conditions, even though the general temporal pattern of GPP dynamics was similar for both. Compared to the low fire severity scenario, high fire severity simulations suggested lower

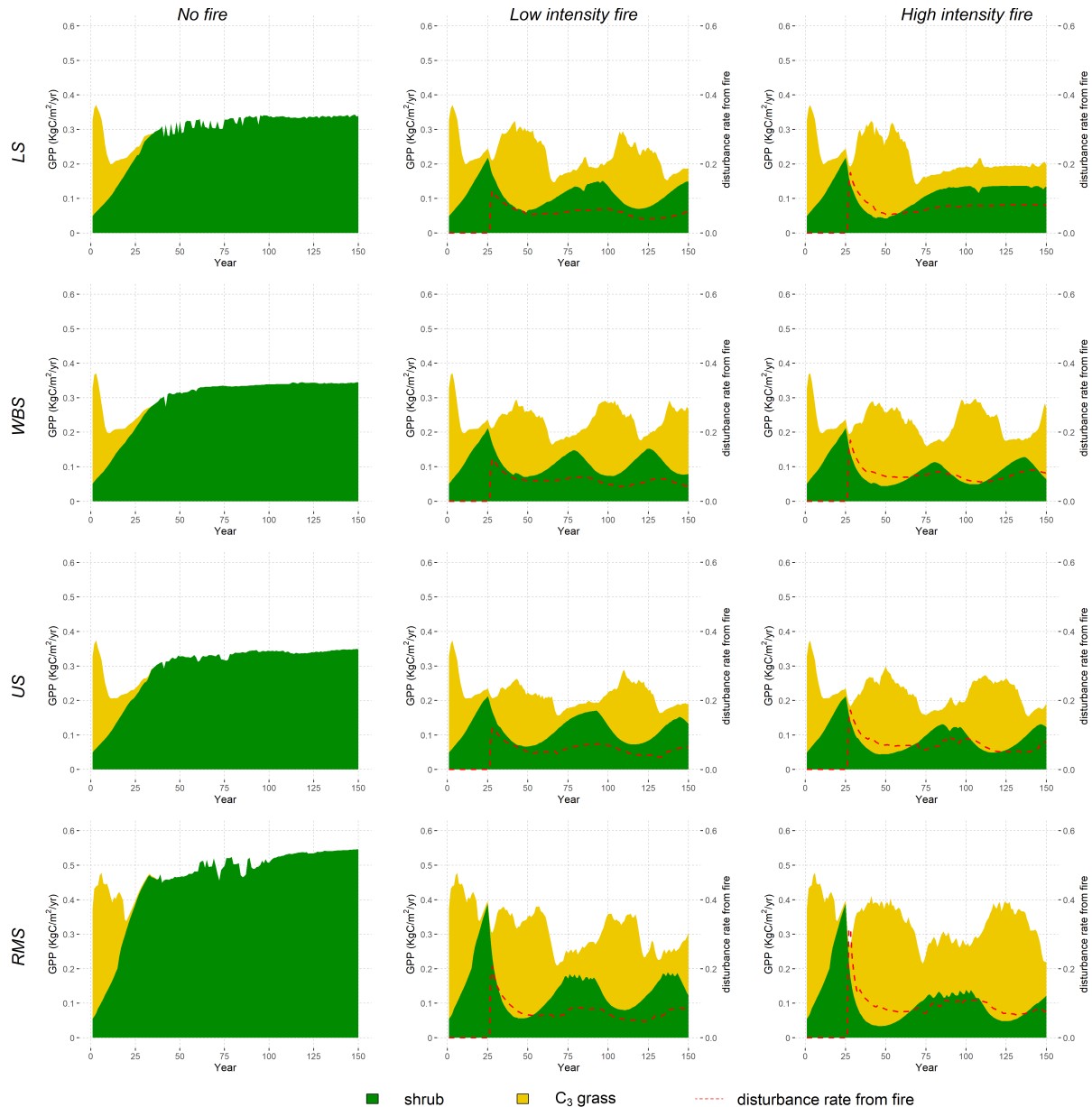

**Figure 2.** Mean annual trends in shrub, $C_3$ grass (temperate $C_3$ grass) and total GPP ($kgCm^{-2}yr^{-1}$) (shrub and $C_3$ grass GPP showed in stack) simulated at four EC flux tower sites (LS, WBS, US, and RMS). Figures in the left column represent the trend in the no fire condition, the middle column the low fire severity condition, and the right column the high fire severity condition. For the model runs with fire conditions, fire was introduced in the 25th year of simulation. The red dashed line is scaled by the secondary y-axis (right), which shows mean fire disturbance rate for the simulation years.

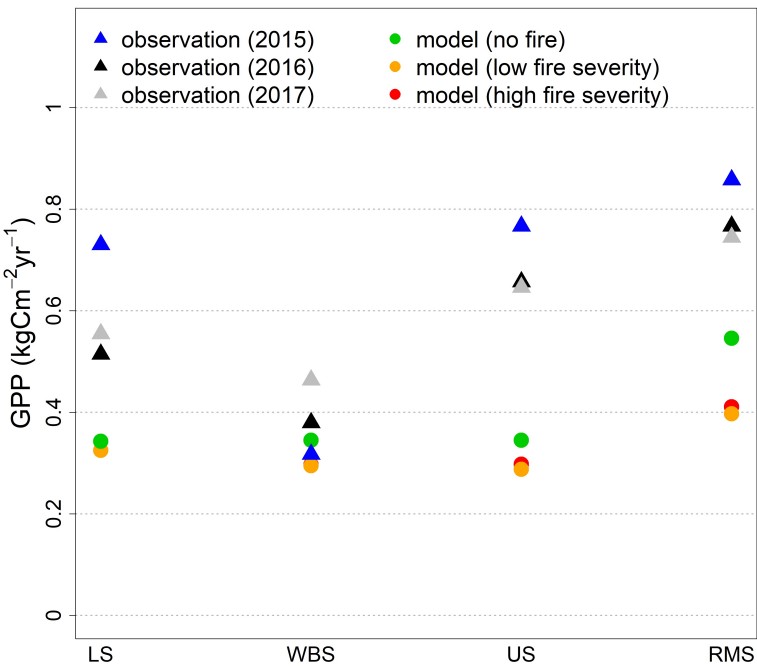

**Figure 3.** Comparison of simulated average annual GPP from EDv2.2 for alternate fire scenarios (no fire, low fire severity, and high fire severity) with observations (from 2015, 2016, 2017) from all four EC tower sites.

peaks of shrub GPP, despite having approximately equal (or even higher for some) levels of total GPP due to higher levels of grass GPP. We can see clear difference in total AGB (Fig. A1) with lower peaks for high fire severity conditions for all four sites. With high fire severity, we observed longer fire return intervals for LS and RMS sites (about 60 years for both LS and RMS) compared to the lower fire severity condition (>100 years for LS and >75 years for RMS). We compared average annual GPP from EDv2.2 for different scenarios (at an equilibrium state for no fire condition and at the peak level for fire conditions) with the observed GPP from EC flux tower sites from 2015, 2016, and 2017 for all four sites (Fig. 3). EDv2.2 underestimated GPP for all sites, with the lowest error for the WBS site ($\approx$12%) and the highest error for the US site ($\approx$ 100%) for the no fire scenario.

## 3.2 Multi-year GPP prediction at regional scale

### 3.2.1 EDv2.2 GPP for fire and no-fire scenarios, Soda Fire scenario

We observed annual variation in GPP predictions for both fire and no-fire scenarios (Fig. 4). Annual variation of GPP in the no-fire model simulation could be mostly attributed to annual climatic variations. Despite the climatic influence, differences

**Table 3.** Pearson's correlation coefficient calculated between modeled GPP and MODIS GPP for burnt, unburnt, and whole area.

| Year | Burnt area | | Unburnt area | | Whole area | |
|------|-----------------------|------------------------------------|-----------------------|------------------------------------|-----------------------|------------------------------------|
| | Number of grids (n) | Pearson's correlation coefficient (r) | Number of grids (n) | Pearson's correlation coefficient (r) | Number of grids (n) | Pearson's correlation coefficient (r) |
| 2015 | 336 | 0.58* | 464 | 0.40* | 800 | 0.50* |
| 2016 | 336 | 0.63* | 464 | 0.46* | 800 | 0.55* |
| 2017 | 336 | 0.57* | 464 | 0.50* | 800 | 0.63* |
| 2018 | 336 | 0.52* | 464 | 0.49* | 800 | 0.63* |
| 2019 | 336 | 0.54* | 464 | 0.55* | 800 | 0.66* |

between fire and no-fire GPP outputs are apparent, especially from 2017 to 2019. High GPP areas in the southwestern regions (in the no-fire simulations) are nearly absent from the fire simulations. The maps in the bottom row of Figure 4 clearly show the differences among the two scenarios. For the first year after fire, there is only a slight reduction in GPP and with no clear spatial pattern. In the second year after fire (2017), GPP was reduced in the fire simulation, at least in some parts (e.g. western region), and shows a clear spatial pattern. From the third year after fire (2018), the reduction in GPP intensified in certain locations while most of the other areas remained similar. In the fourth year (2019), the intensity of GPP reduction got even worse in certain areas while we could also see certain pockets with positive GPP, meaning some recovery for these areas.

We observed obvious differences in EDv2.2 prediction of GPP for shrub PFT and $C_3$ grass PFT for post-fire years (Fig. A2). Since shrub PFT covers the major portion of the overall GPP, the latter is highly influenced by the shrub PFT patterns. While shrub GPP gradually decreased through these years after fire, in contrast, $C_3$ grass started to recover by the third year after the initial reduction in the first and second years (Fig. A2). The pockets of slight recovery in GPP seen in the overall GPP (Fig. 4) appears to be the effect of this $C_3$ grass recovery. These results are in agreement with our results from point-scale fire simulations.

### 3.2.2 EDv2.2 GPP and MODIS GPP, RCEW scenario

Introduction of fire in the northern portion of the study area to the EDv2.2 simulation resulted in observable reduction and recovery of GPP in the burned area (Fig. 5). Modeled GPP reduction in the fire-affected area is a gradual process spanning several years following fire. The first year after the fire showed evidence of some disturbance, however the impact was most evident only during the second (2017) and third years (2018) after fire, based on changes between pre- and post-fire GPP output (Fig. 5). The spatial variation in fire-induced disturbance has close association with elevation (Fig. 1a), which largely influences the precipitation pattern in the study area. Recovery in GPP for the fire-affected area was seen only after the fourth year (2019), even though GPP in the burnt area still lagged behind the unburnt area.

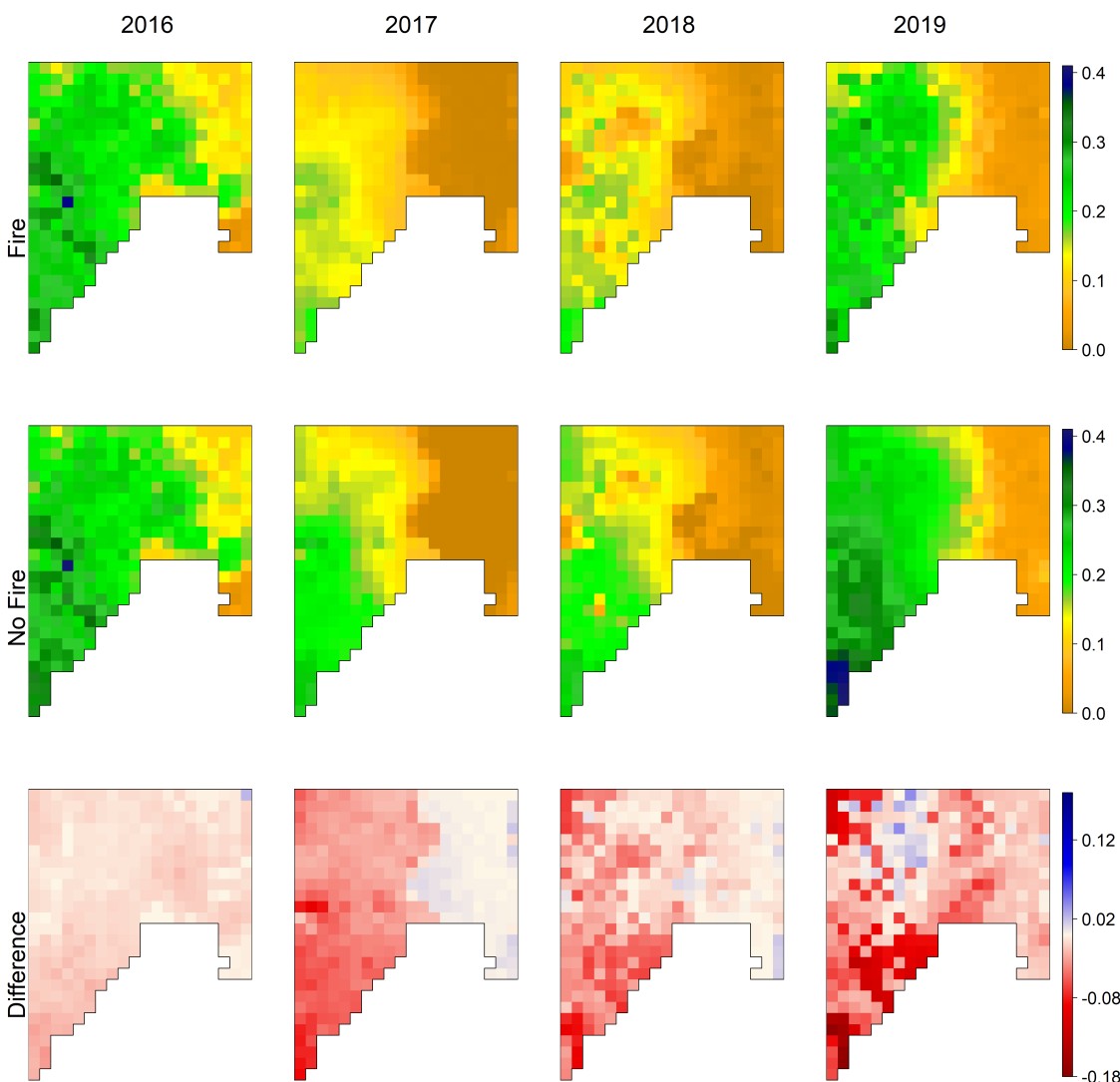

**Figure 4.** EDv2.2 predicted mean monthly GPP $(\mathrm{kgCm^{-2}yr^{-1}})$ for the Soda Fire scenario for July, showing outputs from the model with fire (upper row), without fire (middle row) and difference between two scenarios for the years 2016 to 2019 (representing post-fire years after Soda Fire)

Comparing the pre-fire (2015) EDv2.2 GPP prediction with MODIS GPP revealed an under-prediction across the study area, with major differences towards southern regions (higher elevation areas) of the study area (Fig. 5). The results corroborate our understanding from point-based results where we found better predictions for lower elevation study points compared to those at higher elevations. We observed a clear reduction in EDv2.2 GPP within the fire affected region only the second year after fire (2017), with signs of recovery in 2019. On the other hand, only a slight reduction in MODIS-derived GPP was noted,

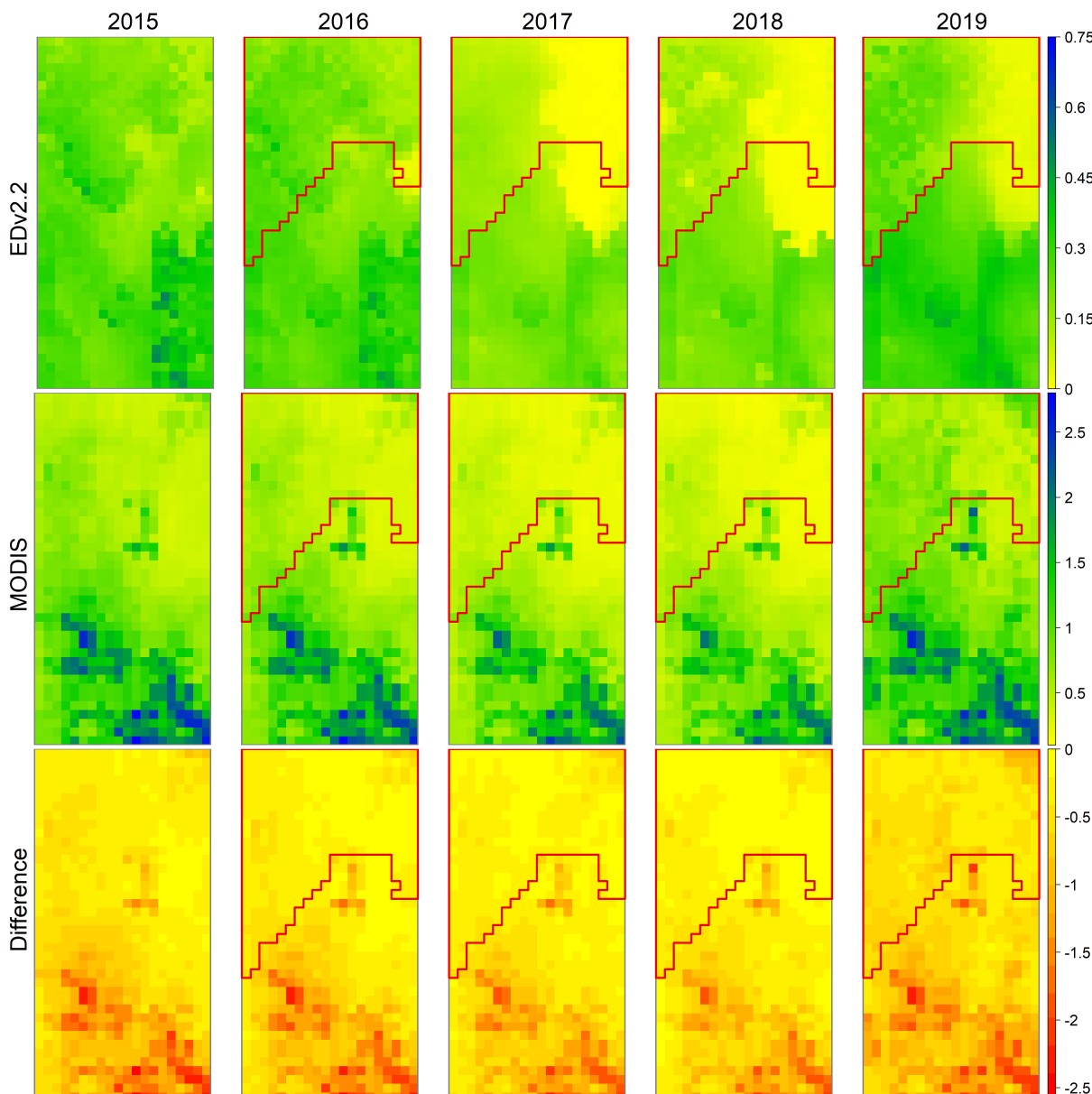

**Figure 5.** Mean monthly GPP ($\mathrm{kgCm^{-2}yr^{-1}}$) for July for the RCEW scenario for the pre-fire (2015) and post-fire (2016 to 2019) years, predicted from EDv2.2 (top-row), derived from MODIS (middle-row) and the difference between two sources (bottom-row). Area surrounded by red polygon represents the area burnt by Soda Fire.

particularly for the years 2017 and 2018, for burnt areas, in the post-fire years. By the year 2019, a good recovery for MODIS GPP was observed.

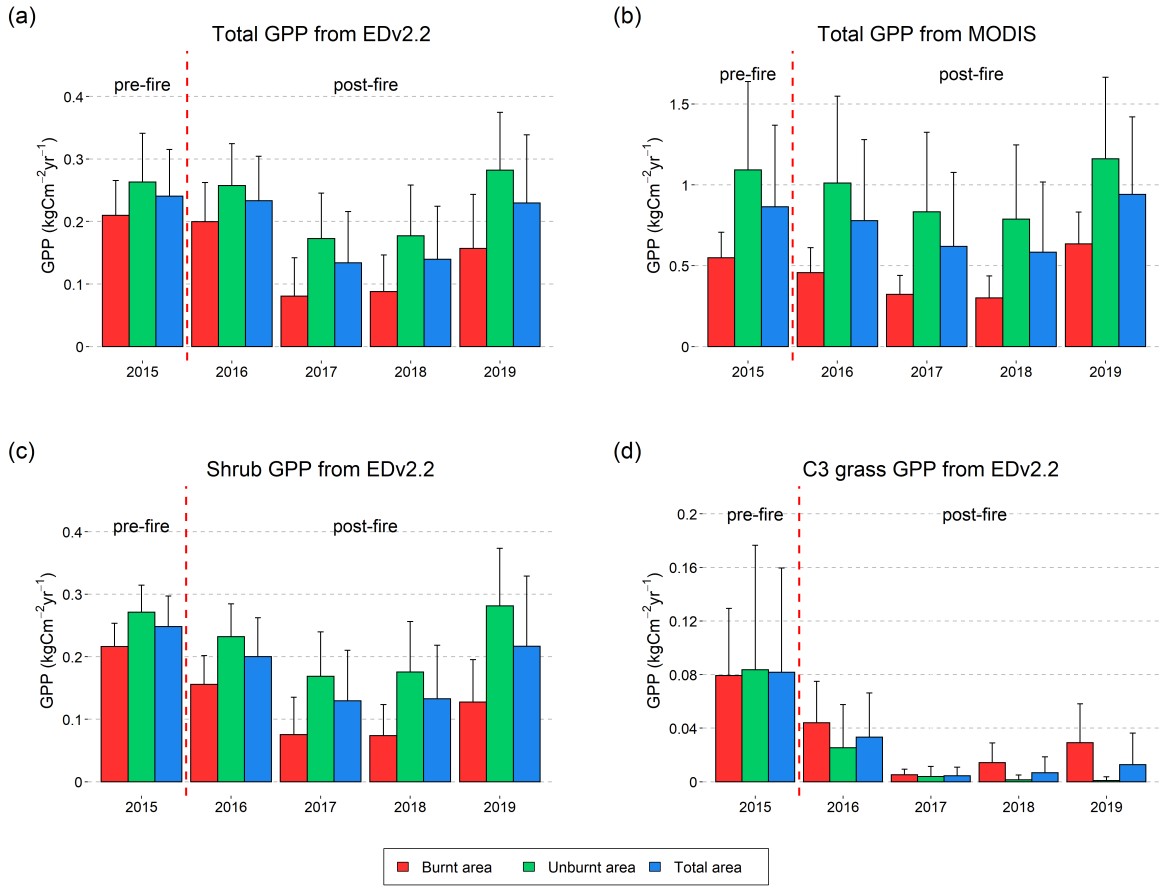

**Figure 6.** Average GPP from EDv2.2 and from MODIS calculated for all the burnt (red), unburnt (green), and total grids (blue) for annual July snapshot maps from 2015 to 2019 (a-b). Error bars in the figure represent ± one standard deviation.

We calculated Pearson's correlation coefficients to further explore the association between modeled GPP and MODIS GPP, which suggested moderate correlations for different areas (Table 3 and Fig. A3). For the entire area and for the unburnt area correlation increased through the years. Weaker correlations for unburnt area in the beginning years (2015 and 2016) could be because of higher variation in vegetation productivity in this area. In contrast, correlation for burnt area slightly increased after
255 fire and dropped back again, revealing more homogeneity and close comparisons immediately after fire.

When mean GPP values from the EDv2.2 simulation and MODIS were plotted for the entire burnt area, unburnt area, and whole area (Fig. 6), there was moderate year-to-year agreement among the two sources in terms of GPP for the entire area. However, there was clear under-prediction of GPP with EDv2.2 compared to that from MODIS, in general. Moreover, while there was not much difference in GPP between burnt and unburnt areas for EDv2.2 in the pre-fire condition, there was already
a huge difference between these areas for MODIS GPP.

EDv.2.2 GPP in the burnt area started to reduce significantly in the second year after fire (2017), continued to remain low until 2018 and showed some recovery in the fourth year. For the modeled GPP, the burnt region had 20% less GPP than unburnt area in the pre-fire year (2015), but this gap changed to 22%, 53%, 50% , and 44% through the first (2016), second (2017), third (2018), and fourth (2019) post-fire years, respectively (Table A1). Though not much variation was observed with MODIS GPP when considering the absolute numbers, as we looked into percent difference in GPP between burnt and unburnt areas, we noticed slight changes through the years. The pre-fire (2015) gap between burnt and unburnt areas for MODIS GPP was 50%, which increased slightly to 55%, 61%, 62% through first, second and third post-fire years, respectively, before reducing this gap to 45% in 2019.

Modeled GPP for shrub followed the pattern of total GPP showing considerable loss in post fire years. One difference with the total GPP was observed during the fourth year after fire, which was prior to shrub recovery. In contrast, we observed different effects on $C_3$ grass GPP. The GPP for $C_3$ grass in burnt areas were slightly higher than unburnt areas immediately after fire in 2016 and showed upward growth trends until 2019. Although, the percent of $C_3$ grass is very low in total GPP, some recovery seen in total GPP in 2019 was primarily associated with the $C_3$ grass growth.

## 4   Discussion

In general, the shrub and grass dynamics modeled in our study are similar to those documented in the literature. With a sustained absence of fire or other disturbance, shrub cover and biomass can dominate over herbaceous species in shrub-steppe ecosystems (Bukowski and Baker, 2013; Cleary et al., 2010; West and Young, 2000), although the complete disappearance of the grass component suggested by our models is unlikely without the influence of other stressors (e.g., livestock grazing).

Thus, this latter dynamic suggests a need for further refinements in PFT development within the EDv2.2 framework, particularly for the $C_3$ grass that we used to represent perennial grasses in the study area. Nevertheless, the EDv2.2 model captures the prevailing trend in ecosystem response to fire, giving it credibility and potential utility as a planning tool. Our modeled fire effects in these ecosystems are also mostly corroborated by the literature in terms of the vegetation loss, PFT competition and recovery. Variation in growth and productivity for $C_3$ grass and shrub after fire disturbance can be understood as their role during different stages of secondary succession. Being an early successional PFT, $C_3$ grass grows quickly and produces high GPP by exploiting favorable growing conditions following the disturbance (Moorcroft et al., 2001). As shrubs start to recover, competition increases at both above and below ground levels for light, water and nutrients, thereby reducing the growth of grass and causing a net loss in total GPP despite an increase in shrub GPP. Most sagebrush species are easily top-killed by fire, do not resprout, and have poor seed viability and dispersal capacity; thus, species of big sagebrush typically require several decades or more to recover to mature conditions post-fire (Baker, 2006; Lesica et al., 2007; Shinneman and McIlroy, 2016). If fire becomes too frequent, shrubs may be prevented from reestablishing, especially in the presence of fire-adapted, nonnative, annual grasses (Brooks et al., 2004). However, even in the presence of nonnative plants, field-based observations suggest that with enough time between fires, shrubs may gradually recover as nonnative herbaceous species dominance declines (Rew and Johnson, 2010; Shinneman and Baker, 2009).

Despite the interannual variability in the observed GPP as evident from the flux tower observation, poor comparisons for the higher elevation sites (US and RMS) than the lower elevation sites (LS and WBS) could be explained by the fact that the shrub parameters we used were mainly developed and calibrated for the lower elevation sites with reasonable agreement (Pandit et al., 2019), and thus may not have accounted for regional variability. Higher ecosystem productivity and quick post-fire recovery at the RMS site compared to the other three sites can be associated with higher site productivity, higher precipitation and lower temperature, as shown in previous studies (Keane et al., 2008; Nelson et al., 2014; Shriver et al., 2018).

With the introduction of fire, we observed drastic change in model predicted GPP values for the burnt area for about 4 years post-fire. An increased reduction in GPP values in burnt area until the third year after fire could be the result of fire behavior in the EDv2.2 model (Longo et al., 2019a), wherein there is a linear increase in burnt area through years given the availability of fuel. There was some recovery in the GPP in the fourth year after fire, mostly because of the increase in $C_3$ grass GPP. Absence of major reduction in MODIS GPP in the burnt area in the post-fire years could be mainly because of perennial grasses and shrubs. Grasses (perennial) could be growing in the second year after fire when conditions are favorable for their growth. The seasonality of the fire also affects how quickly perennial grasses grow back, as a late summer or early fall fire might cause less damage to these grasses (White et al., 2008; Wright and Klemmedson, 1965). A prompt recovery of grass vegetation in the ecosystem was probably not well captured by the EDv2.2 with the default PFT parameters based on a temperate $C_3$ grass.

Fire disturbance phenomena in the EDv2.2 model could not truly represent the true circumstances in the affected area, even though we tried to parameterize the fire severity to match the real scenario. The fire disturbance function in the model did not burn the entire area at once; rather, it selected grids randomly that met the potential fire criteria and killed the vegetation. In addition, this process was gradual and spread over the subsequent years, therefore, we saw the most obvious differences between burnt and unburnt areas until the end of the third year (2018) postfire. Zou et al. (2019) in their study on Region-specific Ecosystem Feedback Fire (RESFire) model with Community Earth System Model also found a decline in GPP until the second year after fire, with a recovery in about eight years. Li et al. (2012) also found similar pattern predicted by CLM-DGVM in burnt areas while testing different fire parameters (Levis et al., 2004; Thonicke et al., 2001) in the model, showing annual variability in burnt area that was at maximum only in the fifth year post-fire. Updating of fire and PFT related parameters along with functional structures for fire-vegetation interactions in the model could better predict burnt areas and vegetation recovery. These findings based on a regional application of a fire module developed explicitly for global applications of a DGVM suggest that future effort is needed to develop more realistic treatments of fire when models like EDv2.2 are applied over smaller regions.

Our GPP outputs from spin-up simulations by EDv2.2 in a near-bare-earth scenario were influenced largely by meteorological forcing data. Our use of modeled meteorological data from the WRF model rather than any field measurements may be an additional source of error. While making these comparisons, we need to consider that there are sources of uncertainty associated with MODIS derived GPP such as mismatching resolutions and limited optimizations (Robinson et al., 2018).

## 5 Conclusions

In this study, we explored fire-induced alterations to GPP in a dryland shrub ecosystem, in terms of shrub and $C_3$ grass PFTs. Results show that the fire model in EDv2.2 captures multi-decadal vegetation dynamics fairly well. While on average the model underestimated GPP compared to flux tower data ($\approx 45\%$), we observed that the model performed well for the lower elevation sites compared to the higher elevation sites. In these simulations, variations due to the elevation gradient were not well captured as the model parameters we used were primarily developed for lower elevation sites. Under the no fire condition, shrubs were dominant and $C_3$ grasses disappeared while approaching an equilibrium state of only shrubs. Simulation results from the WBS site matched well with observations, whereas model results from the remaining three sites underestimated observed GPP data from flux towers. With the introduction of fire, we saw a decline in shrubs and a simultaneous rise in $C_3$ grasses for approximately 3 to 4 decades, followed by slow recovery of shrubs at the expense of grasses. Regional simulation of GPP with EDv2.2 showed continued reduction in GPP for several years post-fire, which only started to increase again with some increase in $C_3$ grass GPP by the fourth year after fire. These modeled GPP trends moderately correlate to what actual GPP trends may be, as indicated by the post-fire GPP response observed from four years of post-fire MODIS imagery.

This study documents an application of EDv2.2 to understand vegetation productivity trends in a semi-arid shrubland ecosystem under alternate fire scenarios at the point scale and provides spatiotemporal trends in vegetation disturbance due to fire disturbance and subsequent recovery at the regional scale. We could reduce uncertainties in comparing model outputs with EC tower observation and satellite-derived products by improving representation of fire and vegetation characteristics and through a more detailed accounting of the errors in input forcing data.

## Appendix A

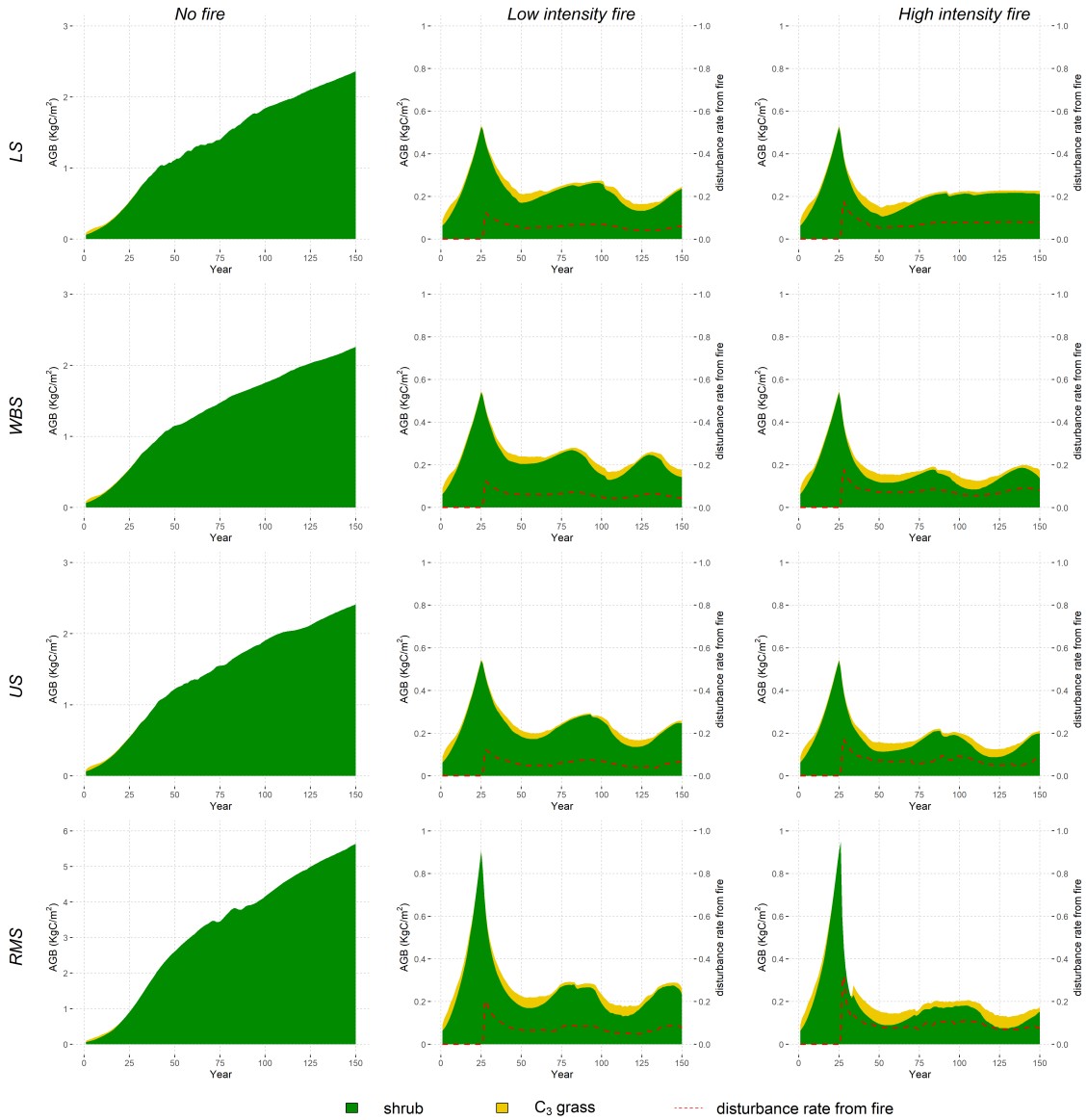

**Figure A1.** Mean annual trends in shrub, $C_3$ grass (temperate $C_3$ grass) and total AGB $(\mathrm{kgCm^{-2}})$ (shrub and $C_3$ grass AGB showed in stack) simulated at four EC flux tower sites (LS, WBS, US, and RMS). Figures in the left column represent the trend in the no fire condition, the middle column the low fire severity condition, and the right column the high fire severity condition. For the model runs with fire conditions, fire was introduced in the 25$^{th}$ year of simulation. The red dashed line is scaled by the secondary y-axis (right), which shows mean fire disturbance rate for the simulation years.

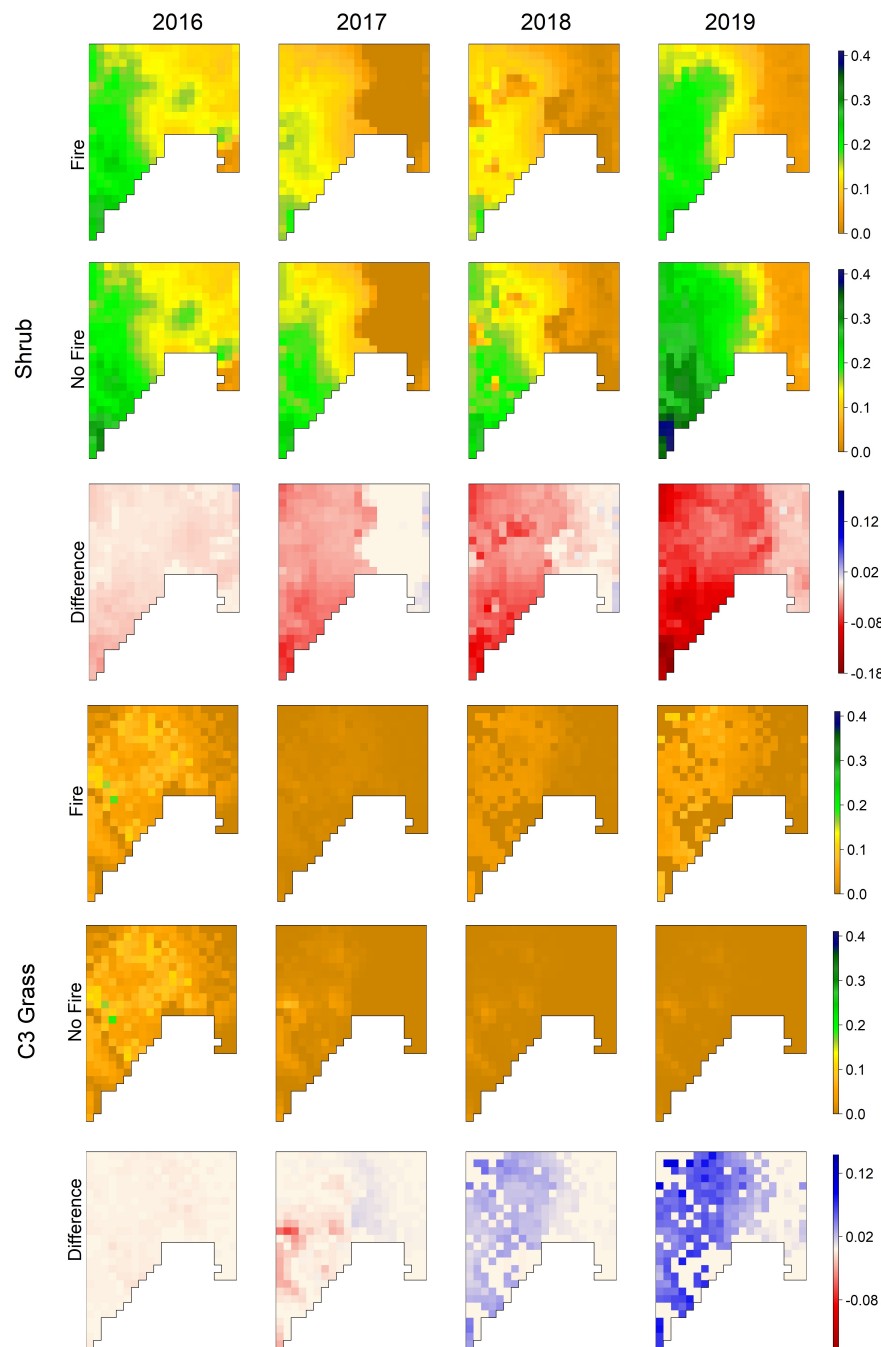

**Figure A2.** EDv2.2 predicted mean monthly GPP $(\mathrm{kgCm}^{-2}\mathrm{yr}^{-1})$ for July for the Soda Fire scenario, separated for shrub and $C_3$ grass PFTs with fire, without fire, and difference between two scenario for the years 2016 to 2019 (representing post-fire years after Soda Fire).

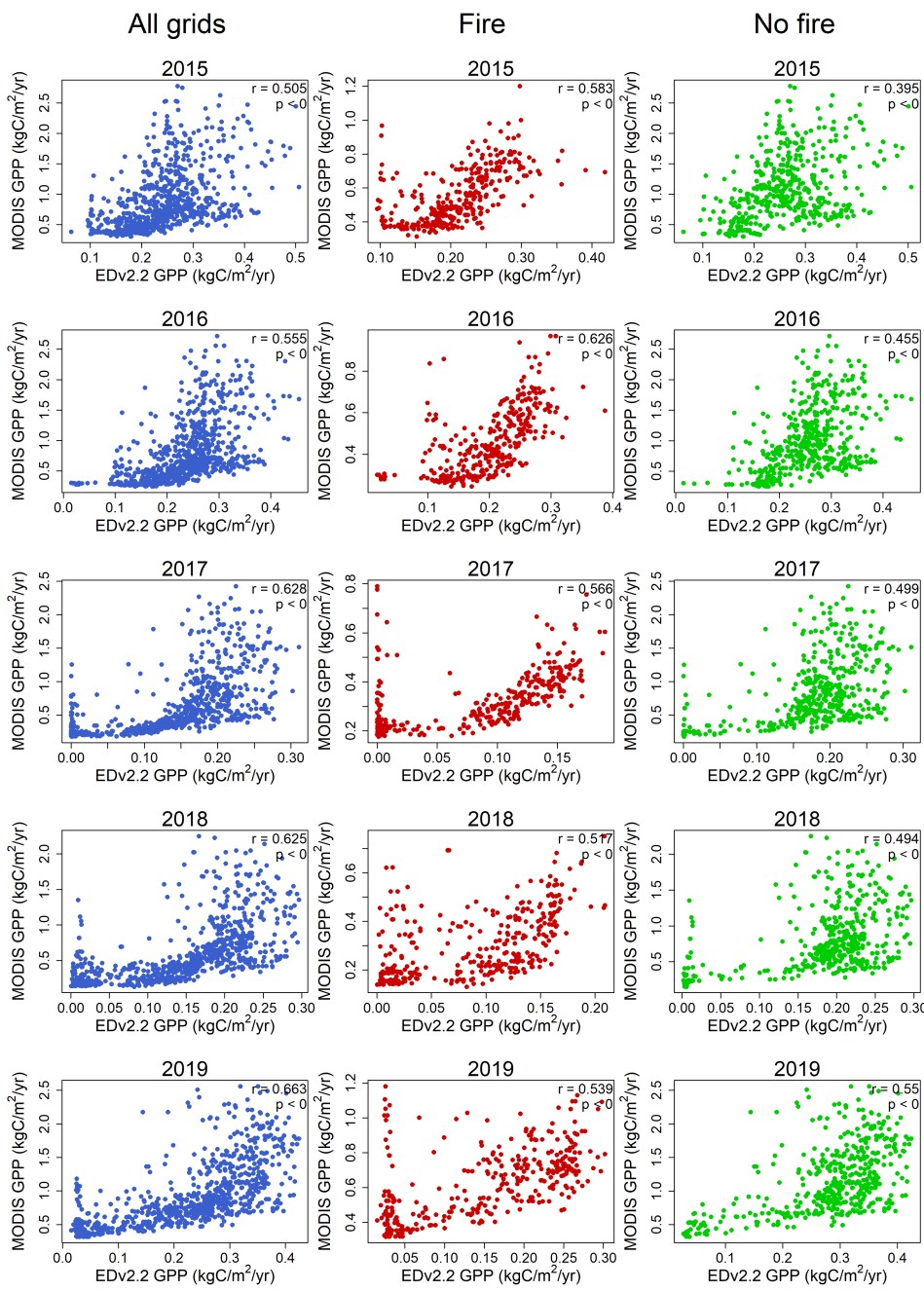

**Figure A3.** Correlation plots between mean monthly GPP $(\mathrm{kgCm^{-2}yr^{-1}})$ values derived from EDv2.2 and MODIS for July of every pre-fire (2015) and post-fire years (2016-2019), categorized by overall, burnt and unburnt grids.

**Table A1.** Percent difference of GPP between burnt and unburnt areas ((GPP in unburnt area - GPP in burnt area)/GPP in unburnt area) for pre-fire and post-fire years.

| Year | MODIS GPP | EDv2.2 GPP (total) | EDv2.2 shrub GPP | EDv2.2 $C_3$ grass GPP |
|------|-----------|--------------------|------------------|------------------------|
| 2015 | 0.50 | 0.20 | 0.20 | 0.05 |
| 2016 | 0.55 | 0.22 | 0.33 | -0.74 |
| 2017 | 0.61 | 0.53 | 0.55 | -0.35 |
| 2018 | 0.62 | 0.50 | 0.58 | -8.71 |
| 2019 | 0.45 | 0.44 | 0.55 | -34.32 |

*Code and data availability.* The original EDv2.2 is available on the GitHub repository at https://github.com/EDmodel/ED2 (ED2 Model Development Team, 2014, last access: 05 November, 2019). EDv2.2 with shrub PFT parameters used in this study is available at https://doi.org/10.5281/zenodo.3461233 (Pandit, 2019, last access: 17 June, 2020), and input data are available at http://doi.org/10.5281/zenodo.4498422 (Pandit, 2021, last access: 04 February, 2021).

*Author contributions.* KP led the model runs and manuscript preparation with significant contributions from all co-authors. KP, HD, ATH,
NFG, ANF and DJS conceived the idea and contributed to the research design.

*Competing interests.* The authors declare that they have no conflict of interest.

*Acknowledgements.* This research has been supported by the NASA Terrestrial Ecology NNX14AD81G, US Forest Service Western Wildlands Environmental Threat Assessment Center (WWETAC) to Boise State University through Joint Venture Agreement (17-JV-11221633-130), and the Joint Fire Science Program Project ID: 15-1-03-23. Field access and support were provided by USDA ARS Northwest Water-
355 shed Research Center. We thank members from Boise Center Aerospace Laboratory (BCAL), Lab for Ecohydrology Alternative Futuring (LEAF) and Research Computing at Boise State University. Part of this work was performed at the R2 computer cluster, Boise State University. Any use of trade, product, or firm names is for descriptive purposes only and does not imply endorsement by the U.S. Government.

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
