# Peer review of "Understanding the effect of fire on vegetation composition and gross primary production in a semi-arid shrubland ecosystem using the Ecosystem Demography (EDv2.2) model"

_Biogeosciences, 2019_

## Referee Comment (RC1) · Anonymous Referee #1 · 25 Feb 2020

Review of Karun et al. : Understanding the effect of fire on vegetation composition and gross primary production in a semi-arid shrubland ecosystem using the Ecosystem Demography (EDv2.2) model

Overview: This study uses a dynamic vegetation model to quantify the impact of fire on GPP in a shrub community. The model is somewhat able to represent observed patterns in vegetation and GPP dynamics after fire. However, I find the manuscript to be somewhat immature, with pieces of the methods section in the introduction, un-satisfying basic description of model parts which are relevant for this study, missing

information in figures etc. and especially a lack of a clear science question or hypotheses to be tested. While I agree that it is worthwhile to improve shrub representation in DGVMs and how these interact with fire, I don't have the feeling the present study takes advantage of the DGVM to ask questions beyond what is known regarding basic impact of fire on sagebrush communities.

Comments

Line 51-71: why would you want to describe the model in this detail in the introduction? This section clearly needs to be moved to the methods. It also needs to be expanded so that one can get a basic idea what the model does, what the fire model does, what happens with the vegetation when a fire occurs etc.

L72-78: Why are you only interested in the effect of fire on GPP, as this is probably the variable where you expect least change through time as vegetation generally is replaced or regrows. In the abstract you mention changes in fire frequency, but you don't follow up on this in your objectives and analysis performed. Probably changes in fire frequency might have an impact, possibly on (soil) carbon, or impact vegetation competition through feedback through the N-cycle, etc. To be clear, I don't say you have to do other analysis, but after reading the manuscript I still wonder why you focused on GPP and no on other aspects of the system which be as relevant.

L 83: Can you give the range in mean temperature and precipitation?

L105: indicate which reanalysis data was used for downscaling using WRF.

L121: Does this mean you don't perform a spinup? How does this work with the N-cycle (which you seem to model, based on what you say in the introduction).

L142: Trends doesn't seem to be the right term, temporal dynamics in GPP? There should exist some literature on vegetation dynamics after fire for these vegetation communities so that you can have an indication whether your simulations capture vegetation dynamics.

L156-157: You don't explain what the driver in the model for this lower GPP with increasing shrub cover is.

L163-164: why didn't you use actual reanalysis forcing so that you can compare interannual variability. Like that one could also assess model performance in figure 4.

L169-170: why? E.g. a fire will burn a shrub immediately, so why would GPP be lowest a couple of years after the fire. When reading this, one wants to know why this happens. Maybe put biomass and GPP for each pft though time in a time series plot or so.

L179-180: I am sorry, but I barely see any difference in delta NDVI between the burned and unburned areas. This is not very convincing, and it almost seems as if there is more of signal from the interannual variability in NDVI due to climate variability then a real fire signal. This entire analysis is a bit shaky; e.g. why do you take GPP for one single day instead of the mean of the month, which should be more representative of hence compare better with NDVI? And possible show the modelled delta GPP between a run with and without fire, instead of comparing between years, so that you only have the fire signal in your simulation results (now one cannot know what is the impact of climate and what is the impact of fire). It would also have taken the mean/median NDVI for multiple images to avoid impact of individual images (especially now that so much Landsat imagery is available).

L212-214: Would have been nice to see a comparison between the model and vegetation dynamics though time as given in the literature.

L 235: I don't understand what you want to say with this sentence.

L234: what do you mean with "annual variability"? I think the discussion needs some work to be more focused and understandable.

Figure 1: include lon-lat and scale to have an idea how big your study area is.

Figure 2: include lon-lat and scale to have an idea how big your study area is. Indicate

what that blob of high NDVI to the northeast is, as it is somewhat distracting.

Figure 3: first sentence of the caption is confusing, shrub, grass and total GPP? Is Grass GPP put on top of shrub GPP?

[Figure]

---

## Referee Comment (RC2) · Anonymous Referee #2 · 3 Mar 2020

General Comments: In this study Pandit et al. aim to understand the effect of fire on vegetation composition and primary production in sagebrush semi-arid ecosystem using a newly developed shrub implementation (Pandit et al., 2019) embedded within EDv2.2. I commend the authors for their addition of a shrub PFT into a DGVM and their work towards better representation of vegetation dynamics in semi-arid systems.

The aims of the study were:

Aim 1: understand the effect of fire on vegetation composition.

Aim 2: understand the effect of fire on primary production.

I have a number of major concerns with respect to this submission. (1) as reviewer 1 pointed out, simulations run to examine how fire affects modelled GPP and compare this with satellite derived NDVI lack a "fire-off" control which uses the same initialisation random seeds, therefore the presented results cannot at this point be attributed to fire effects. These effects could also be due to climate forcing. This lack of control greatly reduces the ability to associate modelled changes in GPP with fire and thus many of the stated results. (2) There is a lack of formal statistical testing on the effect of fire on modelled GPP and fire on NDVI values resulting in a heavy reliance on apparent visual changes being taken as results. I find it necessary that the authors carry out proper significance testing, such testing will greatly improve the manuscript quality.

While the study does attempt to address relevant aims I do not believe they have reached them. There are no concrete conclusions reached in the abstract or discussion which would contribute to understanding the effects of fire on vegetation composition or productivity in semi-arid shrubland systems. Overall this manuscript seems to be more like a model development study than a biogeosciences study.

Specific Comments:

The shrub implementation used by Pandit et al. has already been published in geoscientific model development in 2019, as such I have not gone into detail on the validity of this implementation. Given that the stated aims of the study are to investigate fire effects I found that the lack of proper description of fire in the model greatly impeded my ability to assess the results. Fire apparently affects mortality which is influenced by height (line 69) and on line 124 the two fire severity parameter values used are presented. I am clueless as to how this all works, how fire is distributed across patches, how the shrub implementation influences the probability of mortality, how grasses are treated with respect to fire mortality, and what is fuelling fire. I have no idea what the red line in Fig. 3 (disturbance rate from fire) is showing me.

**BGD**

The bulk of new methods presented appear to have already passed peer review and are presumably valid. Fig. 1 is almost identical to Fig. 1 in Pandit et al. (2019), Table 2 appears to be identical, and large sections of text are very similar to the 2019 paper which is fine for a methods section.

With regard to modelled GPP, GPP appears to be about 50% too low (Fig. 4) apart from at one site, this large discrepancy makes me question whether the approach used is appropriate to understand the effect of fire on GPP. Perhaps I have missed it but the authors only appear to mention this apparent large underestimation on lines 165 and 251 with no further discussion. Please put numbers to this, e.g. GPP at RMS with low fire severity is 50% lower than the observed mean for the 2015-2017 time period. Also the authors should explain why they think the model can appropriately investigate the effect of fire on modelled GPP in spite of these generally rather large underestimations at the plot level.

A major concern with regard to the simulations run to produce Fig. 5, as reviewer 1 pointed out, there is no control simulation run for this area with fire turned off which uses the same initialisation random seeds, therefore the presented results cannot be attributed to fire effects. This lack of control precludes associating modelled changes in GPP with fire and thus many of the stated results, e.g. lines 170-174.

It is puzzling why the authors chose to compare modelled GPP with NDVI. A much better comparison would have been to compare modelled GPP with satellite derived GPP. Indeed, some of the r-squared values from the supplement are very low ($R^2=0.044944$, 2015 unburnt). I am not an expert in satellite derived products but MODIS products appear to be available at the same resolution as simulation runs for the time period. If these data are available simulated GPP should be compared to satellite derived GPP and a control "no-fire" run included.

Overall, a great deal of work needs to be done by the authors in order to allow proper assessment of whether the results are sufficient to support the interpretations. Given

the shown response, or lack thereof, of GPP to fire at the plot level (Fig. 3) and the above mentioned lack of control I remain to be convinced that the changes in GPP presented in Fig. 5 are the result of fire. The lower panel plots in Fig. 5 do not show any clear difference between GPP change in fire vs non-fire areas. In general I would suggest the use of statistical methods to test whether there is a statistically significant difference in GPP between fire and non-fire sites, this would remove the need for eyeballing the results and the need for words such as "suggests" (L172), "hint" (L172), "resembled" (L175), "subtle" (L180). Statistical methods should also be applied to the NDVI changes (NDVI change fire vs no-fire areas) as well as the comparison of GPP change and NDVI change (%change GPP no fire vs %change NDIV no fire) (%change GPP fire areas vs %change NDIV fire areas). I see no signal in the NDVI values which would delineate fire vs no fire areas but proper method can resolve that. Adding a similar satellite derived GPP comparison to modelled GPP, using appropriate statistical methods, would greatly help the authors better make their case.

Minor comments:

L13 + L148 – how do you explain shrub dominance and lack of conifer growth in the absence of fire, shouldn't there be conifer growth in the area which would potentially replace shrubs?

L15 GPP already written out on L10

L21: how are you investigating spatial dynamics? Can fire spread between grid-cells? Perhaps make it more clear what you mean by "spatial behaviour of post-fire ecosystem restoration".

L34: citep(Bradley 2018)

L69: a much better description of fire is needed as commented above.

L99: backslash — (/textitPoa secunda).

L112: table 2. It looks identical to Pandit et al., (2019), not adapted. Perhaps I'm

mistaken.

L147: Off by a decimal place? — 5.0-5.5 kgCm$-$2yr$-$1

L153: it's not clear to me how this fire disturbance works or what the red line is showing. I dont see disturbance following GPP that closely. Why is disturbance highest when shrub GPP is highest rather than when grass GPP is highest? What is fueling the fire? Grass should add a great deal of fuel to the fire yet disturbance is highest when shrub GPP is highest. How often are fires happening?

L158: At LS, why does high fire severity lead to a more stable shrub proportion of GPP?

L162: How do you define stability?

L170: the GPP change 1 year after fire looks to be about the same for the entire study area. why would the biggest change in GPP come two to three years after fire? It's hard to tell whether the changes in GPP are the result of fire or climate.

Table 3: what are the * behind every Pearson number supposed to indicate?

L205: Cite the literature you are referring to.

L212: Cite the literature you are referring to.

L246: "larger contributor to GPP in this ecosystem" citation needed.

---

## Author Comment (AC3) · 21 Apr 2020

Response to interactive comment on "Understanding the effect of fire on vegetation composition and gross primary production in a semi-arid shrubland ecosystem using the Ecosystem Demography (EDv2.2) model" by Karun Pandit et al.

Reviewer 2

**General Comments: In this study Pandit et al. aim to understand the effect of fire on vegetation composition and primary production in sagebrush semi-arid**

**ecosystem using a newly developed shrub implementation (Pandit et al., 2019) embedded within EDv2.2. I commend the authors for their addition of a shrub PFT into a DGVM and their work towards better representation of vegetation dynamics in semi-arid systems. The aims of the study were:**

**Aim 1: understand the effect of fire on vegetation composition.**

**Aim 2: understand the effect of fire on primary production.**

**I have a number of major concerns with respect to this submission. (1) as reviewer 1 pointed out, simulations run to examine how fire affects modelled GPP and compare this with satellite derived NDVI lack a "fire-off" control which uses the same initialisation random seeds, therefore the presented results cannot at this point be attributed to fire effects. These effects could also be due to climate forcing. This lack of control greatly reduces the ability to associate modelled changes in GPP with fire and thus many of the stated results. (2) There is a lack of formal statistical testing on the effect of fire on modelled GPP and fire on NDVI values resulting in a heavy reliance on apparent visual changes being taken as results. I find it necessary that the authors carry out proper significance testing, such testing will greatly improve the manuscript quality.**

**While the study does attempt to address relevant aims I do not believe they have reached them. There are no concrete conclusions reached in the abstract or discussion which would contribute to understanding the effects of fire on vegetation composition or productivity in semi-arid shrubland systems. Overall this manuscript seems to be more like a model development study than a biogeosciences study.**

Thank you for the comments. As per your suggestion and reviewer 1's suggestion, we will run a control simulation (no fire scenario). This will provide a two-way comparison; (i) between control and fire scenario from the model output, and (ii) between fire simulation from the model and satellite observation (MODIS GPP). As per your suggestion,

we will apply a simple t-test to compare burnt and unburnt areas between modelled GPP and MODIS GPP.

Please see our note above to Reviewer 1 about more clearly stating our research questions and subsequent sections for clarity.

**Specific Comments:**

**The shrub implementation used by Pandit et al. has already been published in geoscientific model development in 2019, as such I have not gone into detail on the validity of this implementation. Given that the stated aims of the study are to investigate fire effects I found that the lack of proper description of fire in the model greatly impeded my ability to assess the results. Fire apparently affects mortality which is influenced by height (line 69) and on line 124 the two fire severity parameter values used are presented. I am clueless as to how this all works, how fire is distributed across patches, how the shrub implementation influences the probability of mortality, how grasses are treated with respect to fire mortality, and what is fueling fire. I have no idea what the red line in Fig. 3 (disturbance rate from fire) is showing me.**

As per your comment and reviewer 1's comment we will add a section under methods to elaborate on the model itself and the fire module. We will describe how the fire generates, burns and expands in spatial and temporal terms. In addition, we will include a description of the important parameters that would be influential in causing severe damage and potential recovery for shrub and grasses. The red line in the Fig 3 represents the amount of damage (proportion of grids burnt every year) resulting from fire. It is defined by the available fuel and user selected fire intensity parameter. Available fuel includes all aboveground biomass including grass biomass.

**The bulk of new methods presented appear to have already passed peer review and are presumably valid. Fig. 1 is almost identical to Fig. 1 in Pandit et al. (2019), Table 2 appears to be identical, and large sections of text are very sim-**
**ilar to the 2019 paper which is fine for a methods section. Thank you for your comments. However, in our previous paper (Pandit et al., 2019), from the same study area, we used only two EC tower sites to validate our model. Moreover, in this study, we are only focused on exploring the effect of fire on vegetation dynamics, and we modeled a longer time span. With regard to modelled GPP, GPP appears to be about 50% too low (Fig. 4) apart from at one site, this large discrepancy makes me question whether the approach used is appropriate to understand the effect of fire on GPP. Perhaps I have missed it but the authors only appear to mention this apparent large underestimation on lines 165 and 251 with no further discussion. Please put numbers to this, e.g. GPP at RMS with low fire severity is 50% lower than the observed mean for the 2015-2017 time period. Also the authors should explain why they think the model can appropriately investigate the effect of fire on modelled GPP in spite of these generally rather large underestimations at the plot level.**

We will discuss this issue further. Our primary objective in this study was to understand the effect of fire on vegetation recovery/composition and on primary production . Nevertheless, we will provide justification for such a low model outputs compared with the observation. We will provide numerical comparisons as suggested. We performed our model validation for shrub parameters in our previous study (Pandit et al., 2019), where we benchmarked our model using only two EC tower points (LS and WBS sites), which are at the lower elevation. Results for WBS site is good, and LS is also not very far off. However, the sites RMS and US which were not benchmarked are far off the observation. In our another study which is in review (Dashti et al, in review), we found elevation to be a major factor behind poor model performance for the other sites. Our primary focus in this work was towards understanding the effect of fire by exploring the fire module in the EDv2.2 model by running simulation for different alternate fire scenarios. Our assumption here was we could infer such comparisons using a fairly adapted EDv2.2 model for shrubland based on our previous study.

**A major concern with regard to the simulations run to produce Fig. 5, as reviewer 1 pointed out, there is no control simulation run for this area with fire turned off which uses the same initialisation random seeds, therefore the presented results cannot be attributed to fire effects. This lack of control precludes associating modelled changes in GPP with fire and thus many of the stated results, e.g. lines 170-174.**

As stated above we will perform a no-fire simulation. We hope this will provide better comparison as suggested.

**It is puzzling why the authors chose to compare modelled GPP with NDVI. A much better comparison would have been to compare modelled GPP with satellite derived GPP. Indeed, some of the r-squared values from the supplement are very low (RËĘ2=0.044944, 2015 unburnt). I am not an expert in satellite derived products but MODIS products appear to be available at the same resolution as simulation runs for the time period. If these data are available simulated GPP should be compared to satellite derived GPP and a control "no-fire" run included.**

Thank you for your suggestion. In our revised manuscript, we will compare our model outputs with MODIS GPP information as they are readily available and will make our comparison more justified.

**Overall, a great deal of work needs to be done by the authors in order to allow proper assessment of whether the results are sufficient to support the interpretations. Given the shown response, or lack thereof, of GPP to fire at the plot level (Fig. 3) and the above mentioned lack of control I remain to be convinced that the changes in GPP presented in Fig. 5 are the result of fire. The lower panel plots in Fig. 5 do not show any clear difference between GPP change in fire vs non-fire areas. In general I would suggest the use of statistical methods to test whether there is a statistically significant difference in GPP between fire**

**and non-fire sites, this would remove the need for eyeballing the results and the need for words such as "suggests" (L172), "hint" (L172), "resembled" (L175), "subtle" (L180). Statistical methods should also be applied to the NDVI changes (NDVI change fire vs no-fire areas) as well as the comparison of GPP change and NDVI change (%change GPP no fire vs %change NDIV no fire) (%change GPP fire areas vs %change NDIV fire areas). I see no signal in the NDVI values which would delineate fire vs no fire areas but proper method can resolve that. Adding a similar satellite derived GPP comparison to modelled GPP, using appropriate statistical methods, would greatly help the authors better make their case.**

We observed considerable effects of fire at the plot level as seen in Figure 3. As stated above, we will provide more information on the fire module. We applied average annual meteorological data to remove interannual climate variability, which would otherwise be a major driving factor in GPP simulation. As we kept every other thing constant, and only changed fire parameters, we state that the results in our point simulation are from the fire.

We used similar parameterization, as with point simulation, to run the spatial simulation (Figure 5). However, we used actual annual meteorological data that would allow us to compare with respective years of satellite derived data. We agree that there was little, if any, evidence of fire damage in the NDVI maps. We will explain further in our discussion section the possible conditions that may lead to such situation, including rapid recovery of vegetation (by annual or perennial herbs) as suggested by previous few studies.

As stated above, we will also run a control simulation with no-fire scenario for us to observe and compare between fire and no-fire conditions. In addition, we will apply statistical methods (t-test) to compare model outputs with the satellite observation. As stated above, we will use MODIS GPP instead of Landsat NDVI.

**Minor comments:**

**L13 + L148 – how do you explain shrub dominance and lack of conifer growth in the absence of fire, shouldn't there be conifer growth in the area which would potentially replace shrubs?**

We did not include conifer growth in this study since many of these locations do not have conifers. Moreover, future studies could improve conifer PFTs for local conditions to include in the simulations. We will expand on this in the discussion section.

**L15 GPP already written out on L10**

Thank you. We will correct it.

**L21: how are you investigating spatial dynamics? Can fire spread between grid-cells? Perhaps make it more clear what you mean by "spatial behaviour of post-fire ecosystem restoration".**

We will rephrase the sentence to make it more clear. In this model, although the fire ignition is local it can spread into adjacent grids given favorable conditions such as fuel availability and moisture content. This behavior in interaction with other factors like climate and topography would influence post-fire ecosystem restoration.

**L34: citep(Bradley 2018)**

We will correct it.

**L69: a much better description of fire is needed as commented above.**

As stated above, we will elaborate further on fire module in the EDv2.2 model.

**L99: backslash — (/textitPoa secunda).**

Thank you. We will correct it.

**L112: table 2. It looks identical to Pandit et al., (2019), not adapted. Perhaps I'm mistaken.**

We made minor changes in this table and will make more clear.

**L147: Off by a decimal place? — 5.0-5.5 kgCm−2yr−1**

We will round it to about 5 kgCm-2yr-1.

**L153: it's not clear to me how this fire disturbance works or what the red line is showing. I dont see disturbance following GPP that closely. Why is disturbance highest when shrub GPP is highest rather than when grass GPP is highest? What is fueling the fire? Grass should add a great deal of fuel to the fire yet disturbance is highest when shrub GPP is highest. How often are fires happening?**

We agree that grass may lead to more fuel continuity and hence more frequent and larger fires. However, the fire return intervals may not perfectly align with the most fire-prone fuel conditions. In our model simulations, the disturbance is mainly related to aboveground biomass, so it tends to follow GPP. It appears that under the current model parameterization and structural composition, the shrub GPP has more impact on fuel availability compared to grass, because of the woody nature of the PFT compared to grass and higher biomass storage rates. It looks like the mean fire return interval is somewhat close in length and aligns with peak biomass. Future work on PFT parameterization and fire module would improve the results.

Fire here is an ongoing process after the 25th year. So, the fire related damage will increase when there is available fuel and it will reduce when there is no fuel (aboveground biomass). If we compare this trend as a fire return interval, we can compare this with studies showing fire return periods ranging from 35 years to 435 years for different type of sagebrush ecosystems.

**L158: At LS, why does high fire severity lead to a more stable shrub proportion of GPP? L162: How do you define stability?**

One reason may be that due to high fire damage, shrub has yet to recover before additional fire damage. We defined stability as a long-term maintenance of GPP level

for shrub and grass rather than short fluctuation. We will further explore the literature and revise the sentences to explain this phenomenon.

**L170: the GPP change 1 year after fire looks to be about the same for the entire study area. why would the biggest change in GPP come two to three years after fire? It's hard to tell whether the changes in GPP are the result of fire or climate.**

As described above, disturbance due to fire in the model behaves as a continued process instead of one-time effect. So, there could be spatial growth in fire from one grid to another depending upon fuel and moisture condition. A grid cell not meeting a threshold to get burnt could be burnt next year with slight increase in biomass. In addition, we should definitely take into consideration the effect of climate into these damages. A comparison between fire and no fire scenario, as stated above, may elucidate more details related to this.

**Table 3: what are the * behind every Pearson number supposed to indicate?** They mean significant.

**L205: Cite the literature you are referring to.**

Thank you.

**L212: Cite the literature you are referring to.**

Thank you.

**L246: "larger contributor to GPP in this ecosystem" citation needed.**

Thank you.

---

## Author Comment (AC4) · 24 Apr 2020

Response to interactive comment on "Understanding the effect of fire on vegetation composition and gross primary production in a semi-arid shrubland ecosystem using the Ecosystem Demography (EDv2.2) model" by Karun Pandit et al.

Reviewer 1

**Overview:**

**This study uses a dynamic vegetation model to quantify the impact of fire on GPP in a shrub community. The model is somewhat able to represent observed patterns in vegetation and GPP dynamics after fire. However, I find the manuscript to be somewhat immature, with pieces of the methods section in the introduction, unsatisfying basic description of model parts which are relevant for this study, missing information in figures etc. and especially a lack of a clear science question or hypotheses to be tested. While I agree that it is worthwhile to improve shrub representation in DGVMs and how these interact with fire, I don't have the feeling the present study takes advantage of the DGVM to ask questions beyond what is known regarding basic impact of fire on sagebrush communities.**

Thank you for the comments. We will move the model description from the introduction to methods, rephrase our objective, and clearly state our hypothesis to provide clarity. We will also provide clarity on figures.

We agree there is more work to be done to understand fire in sagebrush communities with EDv2.2 and other DGVMs. However, there is a knowledge gap in understanding the uncertainties of EDv2.2 in assessing the impact of fire on shrub dominated semi-arid ecosystems like the Great Basin region. The aim of this study is to document the potential usefulness and errors in modeling fire behavior with EDv2.2 as a first step in further developing the model for shrublands. Findings from this study has a potential to contribute to substantial utility beyond academic exercise to track shrubland carbon and productivity dynamics at broader scales, as sagebrush is found throughout Western United States and Southwest Canada. Results from our study would also be valuable given this widespread ecosystem is threatened by fire and invasive grasses. Our study could be a preliminary step in that process, to make EDv2.2 a model that can address global changes via dynamics in semiarid shrublands.

We will revise our related sections and include required references to emphasize the importance of this study. In addition, we will add a sentence or two in the conclusion to

re-emphasize these issues and the potential for EDv2.2 to address them with further PFT parameters and fire module refinement.

We will more precisely rewrite our science questions as:

a. What is the projected GPP of a sagebrush-steppe shrub PFT in the Ecosystem Demography (EDv2.2) model for two different fire disturbance scenarios compared against a no-fire scenario (control scenario)?

b. How are the spatial and temporal dynamics of fire disturbance and post-fire recovery represented in EDv2.2 in comparison to remotely sensed data??

**Comments Line 51-71: why would you want to describe the model in this detail in the introduction? This section clearly needs to be moved to the methods. It also needs to be expanded so that one can get a basic idea what the model does, what the fire model does, what happens with the vegetation when a fire occurs etc.**

Thank you for your comments. We will move some of the model description (that is not used as background/introduction) to the methods, and provide additional information on the fire module in a subsection of the Methods.

**L72-78: Why are you only interested in the effect of fire on GPP, as this is probably the variable where you expect least change through time as vegetation generally is replaced or regrows. In the abstract you mention changes in fire frequency, but you don't follow up on this in your objectives and analysis performed. Probably changes in fire frequency might have an impact, possibly on (soil) carbon, or impact vegetation competition through feedback through the N-cycle, etc. To be clear, I don't say you have to do other analysis, but after reading the manuscript I still wonder why you focused on GPP and no on other aspects of the system which be as relevant.**

We used GPP as it is often a direct output of process-based vegetation models. EDv2.2

calculates GPP based maximum photosynthesis using the Farquhar model (Farquhar et al.,1980). In addition, GPP estimates correspond well with remote sensing derived products like NDVI (normalized difference vegetation index), LAI (Leaf area index) and fAPAR (fraction of photosynthetically active radiation absorbed by the vegetation). While we limited our study to GPP, future studies could compare EDv2.2 outputs with remote sensing observations such as net ecosystem production (NEP), leaf area index (LAI), or above ground biomass (AGB). We compared two different levels of fire severity against control (no-fire) scenario at point levels to explore the dynamics of vegetation through the patterns in GPP. The EDv2.2 model could be simulated with alternate N effects including its effect on photosynthesis and decomposition. However, in this analysis we were not focused in the Nitrogen cycle.

**L 83: Can you give the range in mean temperature and precipitation?**

We will provide the range in the revised manuscript. Range of mean annual precipitation range from around 250 to 1100 mm and mean annual temperature ranges from about 5 to 10 °C.

**L105: indicate which reanalysis data was used for downscaling using WRF.**

We used "North American Regional Reanalysis" to downscale WRF data. We will add text and reference to make this clear in the revision (Reference given below).

Reference: National Centers for Environmental Prediction/National Weather Service/NOAA/U.S. Department of Commerce (2005), NCEP North American Regional Reanalysis (NARR), http://rda.ucar.edu/datasets/ds608.0/, Research Data Archive at the National Center for Atmospheric Research, Computational and Information Systems Laboratory, Boulder, Colo. (Updated monthly.) Accessed 01 Nov 2018.

**L121: Does this mean you don't perform a spinup? How does this work with the N-cycle (which you seem to model, based on what you say in the introduction).**

We used existing vegetation state with both shrub and C3 grass to initialize the pointbased simulations. We ran the simulation for 25 years to get the vegetation and other ecosystem conditions such as Nitrogen and soil carbon. However, as suggested in this study we were not focused on assessing N-cycle.

**L142: Trends doesn't seem to be the right term, temporal dynamics in GPP? There should exist some literature on vegetation dynamics after fire for these vegetation communities so that you can have an indication whether your simulations capture vegetation dynamics.**

We agree to rephrase as temporal dynamics in GPP. There are a number of studies assessing GPP recovery and vegetation dynamics after fire. Such studies suggest change in ecosystem carbon exchange from source to sink after fire. These studies show considerable variability in the number of years required to return GPP to pre-fire conditions. Another threat to these ecosystems is that many do not recover and become dominated by exotic annual grass communities that are highly fire prone. We will cite studies which have focused on sagebrush-steppe post-fire recovery, as a comparison to our results. We would also highlight the need for further development of the C3 grass PFTs to better reflect annual grass dynamics in the conclusion section.

**L156-157: You don't explain what the driver in the model for this lower GPP with increasing shrub cover is.**

The main driver behind this dynamic of GPP for two PFTs can be described in terms of secondary succession and competition. In the initial years after fire, there are favorable growing conditions for grasses to grow quickly and produce high GPP. As shrubs start to recover, competition increases, shade is increased and belowground root completion is also higher. These factors reduce the growth of grass thus causing a net loss in total GPP, even with the increase in shrub GPP.

**L163-164: why didn't you use actual reanalysis forcing so that you can compare interannual variability. Like that one could also assess model performance in figure 4.**

We agree that it would be possible to assess model output by comparing results with EC towers if we used respective years of forcing data. However, as the primary intent of this study was to explore the temporal GPP dynamics for two PFTs with fire disturbance as the driving factor, we used an average annual meteorological forcing data and thus minimized interannual variability from weather data. In our previous study on model performance (Pandit et al., 2019), we had applied actual yearly forcing data to perform model validation.

**L169-170: why? E.g. a fire will burn a shrub immediately, so why would GPP be lowest a couple of years after the fire. When reading this, one wants to know why this happens. Maybe put biomass and GPP for each pft though time in a time series plot or so.**

Most vegetation models with fire modules kill plants at different times, which may not correspond to real circumstances. The grids that are not killed (disturbed) in a given year could have higher probability of being killed in the later years as the fuel load increases. Fire damage is also affected largely with the lack of soil moisture in later years. In this analysis we turned on the fire module for post-fire years, which resulted in such a pattern. As suggested, we will revise our figures to show aboveground biomass and GPP for each PFT through time.

**L179-180: I am sorry, but I barely see any difference in delta NDVI between the burned and unburned areas. This is not very convincing, and it almost seems as if there is more of signal from the interannual variability in NDVI due to climate variability then a real fire signal. This entire analysis is a bit shaky; e.g. why do you take GPP for one single day instead of the mean of the month, which should be more representative of hence compare better with NDVI? And possible show the modelled delta GPP between a run with and without fire, instead of comparing between years, so that you only have the fire signal in your simulation results (now one cannot know what is the impact of climate and what is the impact of fire). It would also have taken the mean/median NDVI for multiple im-**

**ages to avoid impact of individual images (especially now that so much Landsat imagery is available).**

We agree that the NDVI maps did not capture equivalent fire damage as suggested by the model and we will adjust this sentence accordingly. In addition, in a semi-arid system like this where moisture limitation is a major driving factor, climate signals could be strong enough to dilute the effects of fire. However, average annual values (Figure 7) do showed some reduction in NDVI from the fire, and the difference in NDVI between burnt and unburnt areas is slightly higher in the first year after fire. In this analysis we were not able to include mean monthly NDVI to compare against the modelled GPP output. We had to limit our comparison for late spring and early summer season when the productivity in the semi-arid ecosystem is sufficiently captured in the satellite images. We used Landsat images with a temporal resolution of 16 days, and there were several occasions when the satellite images were affected with clouds. We selected best images for a year, within the growing season and compared it against GPP output from same day.

In the revised analysis, we will compare our spatial model simulations with MODIS GPP and try to compute mean monthly figures (as we will have several observations in a month) instead of daily figures.

In addition, we will run spatial simulation for a control, ie. no fire condition for the current fire affected area. This will help us show the model behavior more clearly for damage and recovery caused by the fire.

**L212-214: Would have been nice to see a comparison between the model and vegetation dynamics though time as given in the literature.**

We will include this comparison in the discussion.

**L 235: I don't understand what you want to say with this sentence.**

We will rephrase. Our intention is to illustrate results from other studies, where firerelated damage behaved differently compared to satellite observations. As stated earlier, damage defined by these models may lag by a few years depending on biomass and soil moisture conditions.

**L234: what do you mean with "annual variability"? I think the discussion needs some work to be more focused and understandable.**

Thank you for your suggestion. We will revise the discussion for clarity.

**Figure 1: include lon-lat and scale to have an idea how big your study area is.**

Thank you for your suggestion. We will include the lat-long coordinates and scale in the figure.

**Figure 2: include lon-lat and scale to have an idea how big your study area is. Indicate what that blob of high NDVI to the northeast is, as it is somewhat distracting.**

Thank you we will update the figure and provide an explanation of the areas showing higher NDVIs (or MODIS GPPs) in the northeast (that are agricultural and suburban landscapes).

**Figure 3: first sentence of the caption is confusing, shrub, grass and total GPP? Is Grass GPP put on top of shrub GPP?**

Yes, we put grass GPP on top of shrub GPP. Both are stacked and represent a total GPP. We will make this clear in the figure caption.

---

## Author Response (AR1)

Dr. Kirsten Thonicke
Associate Editor
Biogeosciences

Dear Dr. Thonicke,

Thank you very much for your comments on our manuscript. All your comments, along with reviewers' comments were very helpful for us in reshaping and revising this manuscript. We tried our best to revisit the manuscript. Following are responses to your comments/questions on the manuscript. As you understand, this is our preliminary work on fire using EDv.2.2 based on shrubs in dryland ecosystem. We will appreciate your further suggestions/comments on this revision.

Best regards,
Karun Pandit

**Regarding your response to reviewer #1, please**

**1. include your response to the comment on line L156-157 in the manuscript.**
We added some texts on L273-277 to clarify this comment.

**2. make sure you not only add biomass to your figures, but also add the interpretation of those results (GPP change vs. biomass change after) in the manuscript text (reviewer comment to "L169-170: why". You need to thoroughly explain why GPP is reduced over several years after fire and explain it with the model assumptions and model algorithm/equations. This is unusual outcome compared to observations that see a quick GPP recovery after fire.**
We added sentences about fire model behavior in the EDv2.2 in lines L84 ("Area of burnt …") and L90 ("New burnt patches …"). We also added sentences in L291-292 and L301-303 to discuss about the gradual increase in fire damage, and also provided reference (L 305-309) from similar other works that suggest gradual decrease in GPP for initial few years before the vegetation recovery.
We also added two figures to address this concern; one showing temporal pattern of AGB (we can see that AGB is directly related to fire damage in the model) for shrub and C3 PFTs (Fig. A1) in the appendix, and other showing spatial pattern of each PFTs showing post-fire damage and recovery (Fig. A2). We observe C3 grass recovery in three years after fire while shrub GPP is still declining.

**Response to Reviewer 2**
**1. Please add sentences on the conclusion and discussion points of this study in the abstract and check if you have answered (conclusions) and discussed the objectives of your manuscript in the respective sections.**
We added texts at L320-321 to compare our results with the research objective.

**2. Make sure that you address all reviewer comments regarding model description, methods and statistics used in the analysis of model results. The effect of fire on shrub GPP needs to be plausibly described. If you have benchmarked your model already in a previous publication then please cite the outcome in the GPP evaluation.**
We tried to address most of reviewers comments. We have cited and rephrased texts at L284-286 to show results from previous study.

**3. your response to the comment referring to L153: I think your response misses the point raised by the reviewer. Your response explain the general fire ecology in the shrub ecosystem under study, but you need to make sure you explain the curves displaying fire and GPP in your model results. Please check again this point in your model and model analysis and make sure the simulated pattern are sufficiently explained in the manscript.**
We added some text at L191-194 to state that fire damage is directly related to AGB and is also in some way aligned with GPP. We added AGB figure (Fig. A1) at the appendix, to show the trend in AGB for each fire and no fire conditions, and to show corresponding fire damage at the given time.

**4. your response to comment regarding L158: when adding this aspect to your manuscript, please consider the time scales of your simulation study and make sure the stability definition is not flawed by discussing seasonal pattern.**
Thank you for your comment. We removed the term 'stability' as it was a little off-track from our results. We rephrased the sentences at L203-206 to highlight that the difference were more evident for total AGB than GPP, and the fire return interval was longer for some of the sites.

**Response to interactive comment on "Understanding the effect of fire on vegetation composition and gross primary production in a semi-arid shrubland ecosystem using the Ecosystem Demography (EDv2.2) model" by Karun Pandit et al.**

**Reviewer 1**

**Overview:**

**This study uses a dynamic vegetation model to quantify the impact of fire on GPP in a shrub community. The model is somewhat able to represent observed patterns in vegetation and GPP dynamics after fire. However, I find the manuscript to be somewhat immature, with pieces of the methods section in the introduction, unsatisfying basic description of model parts which are relevant for this study, missing information in figures etc. and especially a lack of a clear science question or hypotheses to be tested. While I agree that it is worthwhile to improve shrub representation in DGVMs and how these interact with fire, I don't have the feeling the present study takes advantage of the DGVM to ask questions beyond what is known regarding basic impact of fire on sagebrush communities.**

Thank you for the comments. We moved the model description from the introduction to methods, rephrased our objective, and tried to state our hypothesis with more clarity. We have also reworked on some of the figures to provide clarity on them

We agree there is more work to be done to understand fire in sagebrush communities with EDv2.2 and other DGVMs. However, there is a knowledge gap in understanding the uncertainties of EDv2.2 in assessing the impact of fire on shrub dominated semi-arid ecosystems like the Great Basin region. The aim of this study is to document the potential usefulness and errors in modeling fire behavior with EDv2.2 as a first step in further developing the model for shrublands. Findings from this study has a potential to contribute to substantial utility beyond academic exercise to track shrubland carbon and productivity dynamics at broader scales, as sagebrush is found throughout Western United States and Southwest Canada. Results from our study would also be valuable given this widespread ecosystem is threatened by fire and invasive grasses. Our study could be a preliminary step in that process, to make EDv2.2 a model that can address global changes via dynamics in semiarid shrublands.

We have revised our introduction section and added relevant references to emphasize the importance of this study. In addition, we added texts in the conclusion to re-emphasize these issues and the potential for EDv2.2 to address them with further PFT parameters and fire module refinement.

We have rewritten our science question more precisely as given in the final paragraph of introduction section as given in L65-71.

**Comments**

**Line 51-71: why would you want to describe the model in this detail in the introduction? This section clearly needs to be moved to the methods. It also needs to be expanded so that one can get a basic idea what the model does, what the fire model does, what happens with the vegetation when a fire occurs etc.**

Thank you for your comments. We have moved the major portion of model description from the introduction to the methods and provided additional information on the fire module in a subsection of the Methods at L74-99. We also provided reference to original EDv2.2 model papers that discuss in detail about the model.

**L72-78: Why are you only interested in the effect of fire on GPP, as this is probably the variable where you expect least change through time as vegetation generally is replaced or regrows. In the abstract you mention changes in fire frequency, but you don't follow up on this in your objectives and analysis performed. Probably changes in fire frequency might have an impact, possibly on (soil) carbon, or impact vegetation competition through feedback through the N-cycle, etc. To be clear, I don't say you have to do other analysis, but after reading the manuscript I still wonder why you focused on GPP and no on other aspects of the system which be as relevant.**

We used GPP as it is often a direct output of process-based vegetation models. EDv2.2 calculates GPP based maximum photosynthesis using the Farquhar model (Farquhar et al.,1980). In addition, GPP estimates correspond well with remote sensing derived products like NDVI (normalized difference vegetation index), LAI (Leaf area index) and fAPAR (fraction of photosynthetically active radiation absorbed by the vegetation). While we limited our study to GPP, future studies could compare EDv2.2 outputs with remote sensing observations such as net ecosystem production (NEP), leaf area index (LAI), or above ground biomass (AGB). We compared two different levels of fire severity against control (no-fire) scenario at point levels to explore the dynamics of vegetation through the patterns in GPP. The EDv2.2 model could be simulated with alternate N effects including its effect on photosynthesis and decomposition. However, in this analysis we were not focused in the Nitrogen cycle.

Even though we were not able to redo the analysis with AGB, we tried to add a figure (Fig. A1) showing trend of AGB for different fire and no-fire scenarios. We included reference to AGB along with GPP results at the results section (L184-186; L191-194; L197-199; L201-205)

**L 83: Can you give the range in mean temperature and precipitation?**

We have added texts to provide these information about the study area at L105-106.

**L105: indicate which reanalysis data was used for downscaling using WRF.**

We used "North American Regional Reanalysis" to downscale WRF data. We have added text L131 and reference to make this clear.

**L121: Does this mean you don't perform a spinup? How does this work with the N-cycle (which you seem to model, based on what you say in the introduction).**

We used existing vegetation state with both shrub and C3 grass to initialize the point-based simulations. We ran the simulation for 25 years to get the vegetation and other ecosystem conditions such as Nitrogen and soil carbon. However, as suggested in this study we were not focused on assessing N-cycle.

**L142: Trends doesn't seem to be the right term, temporal dynamics in GPP? There should exist some literature on vegetation dynamics after fire for these vegetation communities so that you can have an indication whether your simulations capture vegetation dynamics.**

We rephrased the term as temporal dynamics in GPP (L139). There are a number of studies assessing GPP recovery and vegetation dynamics after fire. Such studies suggest change in ecosystem carbon exchange from source to sink after fire. These studies show considerable variability in the number of years required to return GPP to pre-fire conditions. Another threat to these ecosystems is that many do not recover and become dominated by exotic annual grass communities that are highly fire prone. We have cited studies related to sagebrush-steppe post-fire recovery, as a comparison to our results (L267; L279-283). We have highlighted the need for further development of the C3 grass PFTs to better reflect annual grass dynamics in the conclusion section (L331-333).

**L156-157: You don't explain what the driver in the model for this lower GPP with increasing shrub cover is.**

The main driver behind this dynamic of GPP for two PFTs can be described in terms of secondary succession and competition. In the initial years after fire, there are favorable growing conditions for grasses to grow quickly and produce high GPP. As shrubs start to recover, competition increases, shade is increased and belowground root completion is also higher. These factors reduce the growth of grass thus causing a net loss in total GPP, even with the increase in shrub GPP. We have added texts on L273-277 to further clarify this comment.

**L163-164: why didn't you use actual reanalysis forcing so that you can compare interannual variability. Like that one could also assess model performance in figure 4.**

We agree that it would be possible to assess model output by comparing results with EC towers if we used respective years of forcing data. However, as the primary intent of this study was to explore the temporal GPP dynamics for two PFTs with fire disturbance as the driving factor, we used an average annual meteorological forcing data and thus minimized interannual variability from weather data. In our previous study on model performance (Pandit et al., 2019), we had applied actual yearly forcing data to perform model validation.

**L169-170: why? E.g. a fire will burn a shrub immediately, so why would GPP be lowest a couple of years after the fire. When reading this, one wants to know why this happens. Maybe put biomass and GPP for each pft though time in a time series plot or so.**

Most vegetation models with fire modules kill plants at different times, which may not correspond to real circumstances. The grids that are not killed (disturbed) in a given year could have higher probability of being killed in the later years as the fuel load (AGB) increases. Fire damage is also affected largely with the lack of soil moisture in later years. In this analysis we turned on the fire module for post-fire years, which resulted in such a pattern. Our comparative analysis between fire and no-fire scenario (regional analysis) shows how the disturbance from fire is in effect until few years after fire.

We added sentences about fire model behavior in the EDv2.2 in lines L84 ("Area of burnt …") and L90 ("New burnt patches …"). We also added sentences in L291-292 and L301-303 to discuss about the gradual increase in fire damage, and also provided reference (L 305-309) from similar other works that suggest gradual decrease in GPP for initial few years before the vegetation recovery.

We also added two figures to address this concern; one showing temporal pattern of AGB (we can see that AGB is directly related to fire damage in the model) for shrub and C3 PFTs (Fig. A1) in the appendix, and other showing spatial pattern of each PFTs showing post-fire damage and recovery (Fig. A2). We observe C3 grass recovery in three years after fire while shrub GPP is still declining.

**L179-180: I am sorry, but I barely see any difference in delta NDVI between the burned and unburned areas. This is not very convincing, and it almost seems as if there is more of signal from the interannual variability in NDVI due to climate variability then a real fire signal. This entire analysis is a bit shaky; e.g. why do you take GPP for one single day instead of the mean of the month, which should be more representative of hence compare better with NDVI? And possible show the modelled delta GPP between a run with and without fire, instead of comparing between years, so that you only have the fire signal in your simulation results (now one cannot know what is the impact of climate and what is the impact of fire). It would also have taken the mean/median NDVI for multiple images to avoid impact of individual images (especially now that so much Landsat imagery is available).**

We agree that the NDVI maps did not capture equivalent fire damage as suggested by the model and we have adjusted this sentence accordingly. In addition, in a semi-arid system like this where moisture limitation is a major driving factor, climate signals could be strong enough to dilute the effects of fire.

We used MODIS derived GPP instead of Landsat NDVI in the revised analysis. In this analysis, instead of making comparison for a given date (a single day), we have made comparisons for July (mean GPP for the month), as described at L173-175.

In addition, we also ran spatial simulation for a control, ie. no fire condition for the current fire affected area as given in L155-157. This helped us show the model behavior more clearly for damage and recovery caused by the fire.

**L212-214: Would have been nice to see a comparison between the model and vegetation dynamics though time as given in the literature.**

We have tried to address this comparison in the discussion such as in L267; L279-283.

**L 235: I don't understand what you want to say with this sentence.**

We apologize for inconvenience. Our intention here was to illustrate results from other studies, where fire-related damage behaved differently compared to satellite observations. As stated earlier, damage defined by these models may lag by a few years depending on biomass and soil moisture conditions.

As stated in previous comment, we have added few sentences as in lines L84, L90, L291-292, L301-303 to discuss about the gradual increase in fire damage in model and provide related reference on similar results.

**L234: what do you mean with "annual variability"? I think the discussion needs some work to be more focused and understandable.**

Thank you for your suggestion. We tried to suggest annual variability between different years of observed data, so we tried to rephrase it to make it more clear in L284.

**Figure 1: include lon-lat and scale to have an idea how big your study area is.**

Thank you for your suggestion. We included lat-lon in one of the maps in study area, which should probably help understand the scope of the study area.

**Figure 2: include lon-lat and scale to have an idea how big your study area is. Indicate what that blob of high NDVI to the northeast is, as it is somewhat distracting.**

We did not include Figure 2 in this analysis as we did not use NDVI for comparison.

**Figure 3: first sentence of the caption is confusing, shrub, grass and total GPP? Is Grass GPP put on top of shrub GPP?**

Yes, we put grass GPP on top of shrub GPP. Both are stacked and represent a total GPP. We have tried to make this clear in the figure caption.

**Reviewer 2**

**General Comments:**

**In this study Pandit et al. aim to understand the effect of fire on vegetation composition and primary production in sagebrush semi-arid ecosystem using a newly developed shrub implementation (Pandit et al., 2019) embedded within EDv2.2. I commend the authors for their addition of a shrub PFT into a DGVM and their work towards better representation of vegetation dynamics in semi-arid systems. The aims of the study were:**

**Aim 1: understand the effect of fire on vegetation composition.**

**Aim 2: understand the effect of fire on primary production.**

**I have a number of major concerns with respect to this submission. (1) as reviewer 1 pointed out, simulations run to examine how fire affects modelled GPP and compare this with satellite derived NDVI lack a "fire-off" control which uses the same initialisation random seeds, therefore the presented results cannot at this point be attributed to fire effects. These effects could also be due to climate forcing. This lack of control greatly reduces the ability to associate modelled changes in GPP with fire and thus many of the stated results. (2) There is a lack of formal statistical testing on the effect of fire on modelled GPP and fire on NDVI values resulting in a heavy reliance on apparent visual changes being taken as results. I find it necessary that the authors carry out proper significance testing, such testing will greatly improve the manuscript quality.**

**While the study does attempt to address relevant aims I do not believe they have reached them. There are no concrete conclusions reached in the abstract or discussion which would contribute to understanding the effects of fire on vegetation composition or productivity in semi-arid shrubland systems. Overall this manuscript seems to be more like a model development study than a biogeosciences study.**

Thank you for the comments. As per your suggestion and reviewer 1's suggestion, we also performed a control simulation (no fire scenario) as given in L154-158. This led us to do a two-way comparison; (i) of EDv2.2 predicted GPP between control (no-fire) and fire scenario, and (ii) between true scenario simulation (including burnt and unburnt areas) from the model and MODIS derived GPP. We could not perform t-test due to major difference in means between model predicted GPP and MODIS derived GPP.

We changed our research questions (L65-71) and subsequent sections as suggested by reviewer #1 for further clarity.

**Specific Comments:**

**The shrub implementation used by Pandit et al. has already been published in geoscientific model development in 2019, as such I have not gone into detail on the validity of this implementation. Given that the stated aims of the study are to investigate fire effects I found that the lack of proper description of fire in the model greatly impeded my ability to assess the results. Fire apparently affects mortality which is influenced by height (line 69) and on line 124 the two fire severity parameter values used are presented. I am clueless as to how this all works, how fire is distributed across patches, how the shrub implementation influences the probability of mortality, how grasses are treated with respect to fire mortality, and what is fueling fire. I have no idea what the red line in Fig. 3 (disturbance rate from fire) is showing me.**

As per your comment and reviewer 1's comment we added texts and a couple of equations under the methods (L74-99) to elaborate on the model itself and the fire module. We tried to summarize the fire related damage. We also included a description of the important parameters that would be influential in causing severe damage and potential recovery for shrub and grasses. The red line in the Fig 3 represents the amount of damage (proportion of grids burnt every year) resulting from fire. It is defined by the available fuel and user selected fire intensity parameter. Available fuel includes all aboveground biomass including grass biomass as given in Equation 1 in L94.

**The bulk of new methods presented appear to have already passed peer review and are presumably valid. Fig. 1 is almost identical to Fig. 1 in Pandit et al. (2019), Table 2 appears to be identical, and large sections of text are very similar to the 2019 paper which is fine for a methods section.**

Thank you for your comments. The table has been slightly adjusted. In our previous paper (Pandit et al., 2019), from the same study area, we used only two EC tower sites to validate our model. The previous study was more about calibration of model using newly derived PFT parameters. In this study, we are only focused on exploring the effect of fire on vegetation dynamics, at extended temporal and spatial scales.

**With regard to modelled GPP, GPP appears to be about 50% too low (Fig. 4) apart from at one site, this large discrepancy makes me question whether the approach used is appropriate to understand the effect of fire on GPP. Perhaps I have missed it but the authors only appear to mention this apparent large underestimation on lines 165 and 251 with no further discussion. Please put numbers to this, e.g. GPP at RMS with low fire severity is 50% lower than the observed mean for the 2015-2017 time period. Also the authors should explain why they think the model can appropriately investigate the effect of fire on modelled GPP in spite of these generally rather large underestimations at the plot level.**

We have tried to discuss this issue further in the manuscript in objectives, results and discussion. We rephrased objective L65-66 to specify that we used parameters from previous study. At L207-208 we rephrased words on comparing EDv2.2 performance. We also touched upon this issue in the discussion at L284-289. However, our objective in this study is to understand the effect of fire on vegetation recovery/composition and on primary production. We performed our model validation for shrub parameters in our previous study (Pandit et al., 2019), where we benchmarked our model using two EC tower points (LS and WBS sites), which are at the lower elevation, with reasonable fidelity. Results from RMS and US which were not benchmarked are far off from the observation. In our another study which is in review (Dashti et al, in review), we found elevation to be a major factor behind poor model performance for the other sites. Our primary focus in this work was towards understanding the effect of fire by exploring the fire module in the EDv2.2 model by running simulation for different alternate fire scenarios. Our assumption here was we could infer such comparisons using a fairly adapted EDv2.2 model for shrubland based on our previous study.

**A major concern with regard to the simulations run to produce Fig. 5, as reviewer 1 pointed out, there is no control simulation run for this area with fire turned off which uses the same initialisation random seeds, therefore the presented results cannot be attributed to fire effects. This lack of control precludes associating modelled changes in GPP with fire and thus many of the stated results, e.g. lines 170-174.**

As stated above we performed a fire/no-fire simulation for a portion of study area (Fig. 4 and Fig. A2) to explore effect of fire against a control (no-fire) simulation (L154-157).

**It is puzzling why the authors chose to compare modelled GPP with NDVI. A much better comparison would have been to compare modelled GPP with satellite derived GPP. Indeed, some of the r-squared values from the supplement are very low (R^2=0.044944, 2015 unburnt). I am not an expert in satellite derived products but MODIS products appear to be available at the same resolution as simulation runs for the time period. If these data are available simulated GPP should be compared to satellite derived GPP and a control "no-fire" run included.**

Thank you for your suggestion. In our revised manuscript, we have compared our model outputs with MODIS GPP information (L157-158). As suggested by reviewer #1, we used monthly mean GPP values from July of each year (from 2015 to 2019) EDv2.2 and MODIS for comparison (L173-175). In addition, we also provided PFT-wise mean change in GPP through years for fire and no-fire areas. We clearly observed higher GPP growth for C3 grass in third and fourth years after fire (both in mean values, Fig6 and spatial maps, Fig A2), while shrub recovery was not evident yet.

**Overall, a great deal of work needs to be done by the authors in order to allow proper assessment of whether the results are sufficient to support the interpretations. Given the shown response, or lack thereof, of GPP to fire at the plot level (Fig. 3) and the above**

**mentioned lack of control I remain to be convinced that the changes in GPP presented in Fig. 5 are the result of fire. The lower panel plots in Fig. 5 do not show any clear difference between GPP change in fire vs non-fire areas. In general I would suggest the use of statistical methods to test whether there is a statistically significant difference in GPP between fire and non-fire sites, this would remove the need for eyeballing the results and the need for words such as "suggests" (L172), "hint" (L172), "resembled" (L175), "subtle" (L180). Statistical methods should also be applied to the NDVI changes (NDVI change fire vs no-fire areas) as well as the comparison of GPP change and NDVI change (%change GPP no fire vs %change NDIV no fire) (%change GPP fire areas vs %change NDIV fire areas). I see no signal in the NDVI values which would delineate fire vs no fire areas but proper method can resolve that. Adding a similar satellite derived GPP comparison to modelled GPP, using appropriate statistical methods, would greatly help the authors better make their case.**

We observed considerable effects of fire at the plot level as seen in Figure 3. As stated above, we have provided further details on the EDv2.2 fire module. We applied average annual meteorological data to remove interannual climate variability, which would otherwise be a major driving factor in GPP simulation. As we kept every other thing constant, and only changed fire parameters, we state that the results in our point simulation are from the fire.

We used similar parameterization, as with point simulation, to run the spatial simulation (Figure 5). However, we used actual annual meteorological data that allowed us to compare with respective years of satellite derived data. Instead of NDVI from Landsat in our previous version, we used GPP from MODIS to make better comparisons. Still there was little, evidence of fire damage observed in the first year after fire. We have tried to explain this in our discussion section the possible conditions that may lead to such situation, including rapid recovery of vegetation (by annual or perennial herbs) as suggested by previous few studies.

As stated above, we also ran a control simulation with no-fire scenario to observe and compare between fire and no-fire conditions. We used MODIS GPP instead of Landsat NDVI for better comparisons.

**Minor comments:**

**L13 + L148 – how do you explain shrub dominance and lack of conifer growth in the absence of fire, shouldn't there be conifer growth in the area which would potentially replace shrubs?**

We did not include conifer growth in this study since many of these locations do not have conifers. Moreover, future studies could improve conifer PFTs for local conditions to include in the simulations. We will expand on this in the discussion section.

**L15 GPP already written out on L10**

Thank you. We corrected it.

**L21: how are you investigating spatial dynamics? Can fire spread between grid-cells? Perhaps make it more clear what you mean by "spatial behaviour of post-fire ecosystem restoration".**

We rephrased the sentence. In this model, although the fire ignition is local it can spread into adjacent patches given favorable conditions such as fuel availability and moisture content. This behavior in interaction with other factors like climate and topography would influence post-fire ecosystem restoration.

**L34: citep(Bradley 2018)**

We corrected it.

**L69: a much better description of fire is needed as commented above.**

As stated above, we have tried to elaborate further on fire module in the EDv2.2 model.

**L99: backslash — (/textitPoa secunda).**

Thank you. We will correct it.

**L112: table 2. It looks identical to Pandit et al., (2019), not adapted. Perhaps I'm mistaken.**

We have made minor changes in this table from the original one.

**L147: Off by a decimal place? — 5.0-5.5 kgCm−2yr−1**

Thank you for the comment. We changed it to 0.55 kgCm-2yr-1.

**L153: it's not clear to me how this fire disturbance works or what the red line is showing. I dont see disturbance following GPP that closely. Why is disturbance highest when shrub GPP is highest rather than when grass GPP is highest? What is fueling the fire? Grass should add a great deal of fuel to the fire yet disturbance is highest when shrub GPP is highest. How often are fires happening?**

We agree that grass may lead to more fuel continuity and hence more frequent and larger fires. However, the fire return intervals may not perfectly align with the most fire-prone fuel conditions. In our model simulations, the disturbance is mainly related to aboveground biomass, so it tends to follow GPP (and more so with AGB). We updated some texts to state what red line is showing and how it is related to biomass. It appears that under the current model parameterization and structural composition, the shrub GPP has more impact on fuel availability compared to grass, because of the woody nature of the PFT compared to grass and higher biomass storage rates. It looks like the mean fire return interval is somewhat close in length and aligns with peak biomass. Future work on PFT parameterization and fire module would improve the results.

Fire here is an ongoing process after the 25$^{th}$ year. So, the fire related damage will increase when there is available fuel and it will reduce when there is no fuel (aboveground biomass). If

we compare this trend as a fire return interval, we can compare this with studies showing fire return periods ranging from 35 years to 435 years for different type of sagebrush ecosystems.

**L158: At LS, why does high fire severity lead to a more stable shrub proportion of GPP? L162: How do you define stability?**

We removed the term 'stability' and showed how high fire severity could actually keep AGB at lower level and for some sites increase fire return interval (as described above).

**L170: the GPP change 1 year after fire looks to be about the same for the entire study area. why would the biggest change in GPP come two to three years after fire? It's hard to tell whether the changes in GPP are the result of fire or climate.**

As described above, disturbance due to fire in the model behaves as a continued process instead of one-time effect. So, there could be spatial growth in fire from one grid to another depending upon fuel and moisture condition. A grid cell not meeting a threshold to get burnt could be burnt next year with slight increase in biomass. In addition, we should definitely take into consideration the effect of climate into these damages. Our updated analysis comparing GPP between fire and no fire scenario, supports the idea that the result should be mostly the result of fire.

**Table 3: what are the * behind every Pearson number supposed to indicate?**

They mean significant.

**L205: Cite the literature you are referring to.**

Thank you.

**L212: Cite the literature you are referring to.**

Thank you.

**L246: "larger contributor to GPP in this ecosystem" citation needed.**

Thank you.

[revised manuscript text omitted]
). Users can choose  one of the two stochastic methods that define fire ignition in the model. Availability of enough fuel (AGB) is the first necessary condition common to both of these methods. The second necessary condition could be set up to be either the total soil water content within a designated depth or the accumulated precipitation for the last 12 months. The fire severity parameter (defined between 0 to 1) in the subroutine determines the level of fire-related disturbance depending upon available fuel. Fire in EDv2.2 affects the vegetation mortality rate, which is a function of cohort height for a given plant function type (PFT). New burnt patches are created every year when the minimum area necessary to generate a new patch is available through the loss of affected cohorts. Disturbance rate from fire ($\lambda_{\mu,\mu_0}^{FR}$) for a patch $u$ (given by subscript $u$) is given by the following equation (Eq. 1) as originally defined by Moorcroft et al. (2001) and later revisited by Longo et al. (2019b).

~~In this study, we used the version of EDv2.2 with shrub PFT (Pandit et al., 2019) to understand the effect of fire on a shrubland ecosystem in the Reynolds Creek Experimental Watershed (RCEW) , Great Basin, USA. We explored the dynamics of shrub and C3 grass gross primary production (GPP) under alternative fire scenarios for four different sites (point-based analysis) in the study area, and also investigated model generated post-fire shrubland recovery patterns against observed Landsat image-derived Normalized Difference Vegetation Index (NDVI) (Wylie et al., 2003; Running et al., 2004) covering the entire RCEW area (regional-based analysis)~~

$$\lambda_{\mu,\mu_0}^{FR} = I \sum_{u=1}^{N_p} \sum_{k=1}^{N_{T_u}} \left\{ \left[ C_{ul_k} + F_{AG_{uk}}(C_{u\sigma_k} + C_{uh_k}) \right] \gamma_u \alpha_u \right\} \tag{1}$$

where $N_p$ is number of patch, $N_{T_u}$ is number of cohort in patch u, $\gamma_u$ is binary ignition function which is defined as given in equation 2 below, $\alpha_u$ is relative ara of patch, 
[revised manuscript text omitted]

250  clear spatial pattern. In the second year, loss of GPP from fire is clearly increased, at least in some parts (western region), and shows a clear spatial pattern. From the third year, loss of GPP intensifies in certain locations while most of other areas remain similar. In the fourth year, the intensity of loss even gets worse in certain areas, and we can also see certain pockets with positive GPPs, meaning some recovery for limited areas.

We observed obvious difference in EDv2.2 prediction of GPP for shrub PFT and C3 grass PFT for post-fire years (Fig. A1).
255  Since shrub PFT covers major portion of the overall GPP, the later is highly influenced by the pattern of shrub PFT. While

**Table 3.** Pearson's correlation coefficient calculated between modeled GPP and  MODIS GPP for burnt, unburnt, and whole area.

| Year | Burnt area | | Unburnt area | | Whole area | |
|------|-----------|--------------------------------|--------------|--------------------------------|------------|--------------------------------|
| | Number of grids (n) | Pearson's correlation coefficient (r) | Number of grids (n) | Pearson's correlation coefficient (r) | Number of grids (n) | Pearson's correlation coefficient (r) |
| 2015 | 336 | 0.58* | 464 | 0.40* | 800 | 0.50* |
| 2016 | 336 | 0.63* | 464 | 0.46* | 800 | 0.55* |
| 2017 | 336 | 0.57* | 464 | 0.50* | 800 | 0.63* |
| 2018 | 336 | 0.52* | 464 | 0.49* | 800 | 0.63* |
| 2019 | 336 | 0.54* | 464 | 0.55* | 800 | 0.66* |

shrub GPP is gradually decreasing in through these years after fire, in contrast, $C_3$ grass starts to recover by third year after initial loss in the first and second year (Fig. A1). The pockets of slight increase in GPP seen in overall GPP (Fig. 4) appears to be the effect of this $C_3$ grass recovery. These results are in agreement with our results from point-scale fire simulations.

**3.2.2 EDv2.2 GPP and MODIS GPP**

260 Introduction of fire in the northern portion of the study area to the EDv2.2 simulation resulted in observable loss and recovery of GPP in the burned area  (Fig. 5). Modeled loss of GPP in the fire-affected area is a gradual process spanning several years following fire  . The first year after the fire showed evidence of some disturbance, however the impact was most evident only during the second (2017) and third years (2018) after fire, based on changes between pre- and post-fire GPP output (Fig. 5). The spatial variation in fire-induced disturbance  has close association with elevation (Fig. 1), which largely

265 influences the precipitation pattern in the study area.  Recovery in GPP for the fire-affected area is seen only after the fourth year (2019), even though GPP in the burnt area still lags behind the unburnt areas.

 When we compared pre-fire  , along with some notable geographic differences. The  (2015) EDv2.2

270  GPP prediction with MODIS GPP, we found that there is an under-prediction across the study area,  with major differences towards southern region (higher elevation areas) of the study area (Fig. 6). The results corroborates with our understanding from point-based results where we found better predictions for lower elevation study points compared to those at higher elevations. We can observe a clear

[Figure]

**Figure 4.**  EDv2.2 predicted mean monthly GPP ($kgCm^{-2}yr^{-1}$) for the month July,   showing outputs from the   model with fire (upper row), without fire (middle row) and difference between two scenario for the  years   2016 to 2019 (representing post-fire years after Soda Fire)

275  reduction in EDv2.2

[Figure]

**Figure 5.**  Mean monthly GPP ($kgCm^{-2}yr^{-1}$) for the month of July for entire study area 
[revised manuscript text omitted]
. Results show that fire model in EDv2.2 capture long-term vegetation dynamics fairly well while fire model behavior resulted in mismatch at short-term predictions when compared with MODIS GPP. Under the no fire condition, shrubs were dominant and C3 grasses disappeared while approaching an equilibrium state of pure shrubs. Simulation results from the WBS site matched well with observations, whereas model results from the remaining three sites underestimated observed GPP data from flux towers. With the introduction of fire, we saw a decline in shrubs and a simultaneous rise in C3 grasses for approximately

3 to 4 decades of time, followed by slow recovery of shrubs at the expense of grasses. Regional simulation of GPP with EDv2.2 showed continued reduction in GPP for several years post-fire, which only started to increase again with  some increase in C3 grass GPP by the fourth year post-fire. These modeled GPP trends moderately correlate to what actual GPP trends may be, as indicated by the post-fire  GPP response observed from four years of post-fire  MODIS imagery.

This study documents an application of EDv2.2 to understand vegetation productivity trends in a semi-arid shrubland ecosystem under alternate fire scenarios at the point scale and  provides spatiotemporal trends in vegetation disturbance due to fire disturbance and  subsequent recovery at the regional scale. We could reduce uncertainties in comparing model outputs with EC tower observation and satellite-derived products by improving representation of fire and vegetation  characteristics and through a more detailed accounting of the errors in input forcing data.

**Appendix A**

[Figure]

**Figure A1.** Mean annual trends in shrub, $C_3$ grass (temperate $C_3$ grass) and total AGB ($\mathrm{kgCm^{-2}}$) (shrub and $C_3$ grass AGB showed in stack) simulated at four EC flux tower sites (LS, WBS, US, and RMS). Figures in the left column represents the trend in the no fire condition, the middle column the low fire severity condition, and the right column the high fire severity condition. For the model runs with fire conditions, fire was introduced in the 25[th] year of simulation. The red dashed line is scaled by the secondary y-axis (right), which shows mean fire disturbance rate for the simulation years.

[Figure]

**Figure A2.** Mean monthly GPP ($\mathrm{kgCm^{-2}yr^{-1}}$) for the month July of every year . The starting year (2015) shows the pre-fire condition in the $25^{th}$ year of spin-up, and the 4 subsequent years represent annual conditions after fire. Maps in the bottom show change in GPP every subsequent year after the fire incident compared to the pre-fire condition in 2015.

[Figure]

**Figure A3.** Mean monthly GPP $(\text{kgCm}^{-2}\text{yr}^{-1})$ for the month July of every year . 
[revised manuscript text omitted]

---

## Author Response (AR2)

January 4, 2021

Dr. Kirsten Thonicke
Associate Editor
Biogeosciences

Dear Dr. Thonicke,

Please accept our revised manuscript *"Understanding the effect of fire on vegetation composition and gross primary production in a semi-arid shrubland ecosystem using the Ecosystem Demography (EDv2.2) model"* for consideration for publication in Biogeosciences.

In this revised manuscript, we have addressed your comments. We have uploaded a response to comments, track-changes version of the manuscript and a clean version of the manuscript. Please let us know if you have any further comments or instructions regarding the manuscript.

Thank you.

Sincerely,
Karun Pandit
Corresponding author
Postdoctoral researcher
University of Florida
karunpandit@gmail.com
315 708 3901

*Dear Dr. Thonicke,*
*Thank you so much for your time and efforts in providing these valuable comments which have helped us to improve this manuscript. We have tried to address your comments point by point carefully. Please let us know if we misunderstood any of your comments or instructions.*

*Our responses are provided in blue fonts. Line numbers we have referred in the response correspond to tracked changed version of the revised manuscript.*

Associate Editor Decision: Reconsider after major revisions (09 Nov 2020) by Kirsten Thonicke

Comments to the Author:

Dear Dr. Pandit,
thank you very much for your revisions and detailed responses. Your manuscript has advanced but still requires further improvements or explanations.

One major issue that I still have is that you write in line 285 that the simulated pattern might be influenced by the model not being in equilibrium. I think this is a crucial point. If the simulated fire and GPP pattern that you analyse in your manuscript are additionally influenced by the model getting into equilibrium then it is unclear if the pattern you describe are solely driven by climate and you can draw the conclusion as you did. I apologize for not spotting this point earlier, but I regard this as a crucial point which I ask you to cross-check in your modelling protocol, if your model results are in equilibrium with climate and you then only simulate the influence of transient climate on fire and GPP. Because this point might require more time to revise and adjust the results shown in the manuscript, I will reconsider your manuscript after major revisions (it is more work than minor revisions).

*Thank you for your important comment.*

*The mean annual GPP for the entire study area is in equilibrium after ~ 20 years of model runs (see figure below). After 25$^{th}$ year (in 2015) of near-bare-earth simulation, we ran the model further with fire experiments. The figure below shows mean annual GPP beginning first year after simulation to few years beyond 2015 with no fire scenario for the entire study area.*

*We understand the number of years applied for spin-up simulation is limited compared to the number of years commonly suggested for DGVMs to reach equilibrium. We should note that drylands are generally low productive ecosystems, and it is expected that productivity measures such as GPP reach equilibrium in a relatively short time. In previous studies using the Ecosystem Demography (ED) model, even in more productive ecosystems such as forests, productivity variables reached equilibrium rather quickly with similar vegetation*

initialization (0.1 plants/m$^2$) as ours. For example,  Hurtt et al (2004) has shown that LAI reached its peak at around 20 years of model simulation whereas Moorcroft et al., 2001 suggested that the biomass growth rate drops sharply after about 30-50 years of simulation.

We acknowledge that 20 years of spin up may not be enough for some slow processes such as soil carbon pools which might require hundreds to thousands of years to reach an equilibrium. Indeed, over hundreds of years, fire ultimately changes the soil carbon pool and hence GPP through different processes such as a change in biome distribution, microbial composition, lateral carbon transport, etc. (e.g. Calvo et al, 2015; Chapin III et al,  2009). Such multi-century analyses are out of the scope of our current study as we focussed on multi-year and multi-decadal relationships between fire and GPP. We made the assumption that the effect of slow processes that affect soil carbon pools on the GPP-fire relationship can be ignored. Following this consideration on equilibrium, we removed the sentence in line 266 "… the simulated pattern might be influenced by the model not being in equilibrium."

We also added the following text in the quotation mark at L 173-176 and Figure 1 below in the supplement.

"To perform these simulations, we initialized EDv2.2 with a near-bare-earth scenario of 0.1 plants m$^2$ for all allowed PFTs (i.e. C$_3$ grass, shrub, northern pines and late conifers) from 1990 and ran it for the following 25 years. Our analysis indicated that 25 years of spin-up was sufficient for GPP to reach equilibrium."

[Figure]

Figure 1. Mean annual gross primary productivity (GPP) averaged over the region from 1991 (first year after simulation) to 2022 based on spin-up from near-bare-earth (0.1 plants/m$^2$) vegetation initialization

In addition to this point I have the following points which I ask you to implement in the revised version:

Abstract/ Introduction
Line 4: correct to Dynamic Global Vegetation Model to come up with the abbreviation DGVM.

*Thank you for pointing this out, this has been corrected for DGVM.*

Line 11: Why EDv2.2 is described as "dynamic vegetation model" not DGVM? If only "dynamic vegetation model" is meant, then correct the justification for a DGVM in the sentences before and justify dynamic vegetation models instead. This also applies to your respective paragraphs in the introduction (lines 48-69).

*Thank you, corrected here and checked throughout for consistency.*

Methods
Line 89: delete "about the model"

*Removed*

Line 111-112: add meaning of μ and μ0 for λ, correct to "relative area"

*We added the meaning of above terms in the equation.*

Line 115: decription and explanation of terms in equation incomplete and unclear, please correct. Explain in a separate sentence what the basic idea of equ. 2 is. Why does soil depth play a role in fire ignition? This is totally unclear. Even if it has been described in a previous paper, this is an important feature of the model that the reader of this manuscript needs to fully understand.

*Thank you. We have elaborated on equation 2 and the role of soil depth in calculating soil dryness.*

Study area
Use "RCEW watershed" throughout the manuscript when you refer to the regional simulations, and "soda fire" when you refer to the Soda fire regional simulations. This way you make the use of these terms consistent with what is shown in Fig. 1.

*We clarified this in Section 2.5 (as Soda Fire scenario and RCEW scenario), and then referred to these scenarios in the subheadings of 3.2.1 and 3.2.2, as well as in Figures 4 and 5 captions.*

Caption Fig. 1: Add the information that the grey area in the large map marks the extent of the Great Basin.

*Added.*

Line 125: define here, clearly in the text, that you have another definition of your study area to run your regional simulations.

*We clarified this in the Study Area section, referring to Figure 1b to demonstrate the study areas of the regional simulations.*

Line 205: replace "making comparisons with mean monthly GPP of July from the model." with "comparing them against simulated mean monthly GPP values of July."

*Corrected.*

Methods and results:
1) replace "long-term" with decadal or "mult-year". "Long-term" is unspecific and people have different understanding what "long-term" stands for. Use "seasonal" instead of "short-term", again this is a unspecific description of time scale.

*Thank you for the suggestion. We replaced long-term with multi-decadal and short-term with multi-year.*

2) Use subscripts of C3 and CO2 throughout the manuscript consistently.

*Corrected to $C_3$ throughout the manuscript.*

Results
Line 241: it is "underestimated" not "underpredicted"

*Corrected.*

Fig. 2: the fire model simulates fire fractions every year. Is this realistic (how does it compare to observed data) and how can it be explained by the design of the fire model (main assumptions, refer to equ. 1 and 2).

*When the fire sub-module in EDv2.2 is turned on it leads to disturbances in a continued manner (at annual steps) depending upon the availability of fuel (aboveground biomass) and soil dryness as given in Eq. 1 and Eq.2. This model assumption is useful in performing multi-decadal or multi-century analyses for ecosystems which are frequently affected with fire, such as dryland ecosystems in the Western U.S. In such ecosystems, with the fire module on, the EDv2.2 model should be able to closely reproduce vegetation conditions similar to observations. In our multi-decadal point-based analysis we observed regrowth and co-existence of shrub and $C_3$ grass (closely resembling the ecosystems) that what was simulated with fire on. In addition, this assumption also helps us analyse approximate fire return intervals (as revealed by peak disturbance rates in Fig. 2). However, such assumptions may not work well if we are doing simulations for a limited number of years.*

*We have added the following text to further elaborate the assumptions of the model at L91-96.*

*"Along with other disturbance factors in EDv2.2, the fire sub-module creates and maintains age- and size-based heterogeneity at sub-grid levels to closely resemble a broad range of structure and composition in a disturbed ecosystem. For example, a study from South America by Longo et al. (2019a) showed that this model represented a fire disturbed ecosystem like woody savanna very well."*

Line 245 ff and throughout the section: Please rewrite "loss of GPP" think of recovering GPP instead, Specify which row in Fig. 4 you refer to, because Fig. 4 actually shows a comparison between the fire and no-fire experiments. GPP is low, and still less than the initial conditions in 2016. In 2019 this spatial pattern is still different from 2016. This description is missing in your results paragraph. It is not "loss", it is rather "reduced GPP".

*Corrected.*

Line 255: replace "later" with "latter"

*Corrected.*

Line 270: Check if corrected sentence is complete.

*Line 270 appears OK, please let us know if we misunderstood.*

Line 285: if your transient simulations have not yet reached an equilibrium, then please revise your modelling protocol accordingly, and re-run the simulations so that the simulation results are a result of climate and fire and not modelling artefacts. This is crucial for the manuscript to get published!

*Please refer to our previous response and figure.*

Line 299: delete remaining part of an old sentence: "For EDv2.2 GPP"

*Deleted*

Caption of Figure 6: explain colors

*Corrected*

Discussion
262: Forkel et al. 2019 is a global study. It is not comparable to the spatial and temporal scale of your study, please delete.

*This has been deleted, thank you for the correction.*

Line 374: if you see potential errors occurring from WRF, specify which those could be and how do they influence your results. Same applies for MODIS GPP data (line 379).

*We elaborated at lines L 338-341. We stated that there is an additional source of error in using WRF given it is a modelled product compared to potentially using field meteorological data. Likewise, we cited some of the sources of uncertainties associated with MODIS derived GPP.*

*"Our use of modeled meteorological data from the WRF model rather than any field measurements may be an additional source of error. While making these comparisons, we understand that there are also sources of uncertainty associated with MODIS derived GPP such as mismatching resolutions and limited optimizations (Robinson et al., 2018)."*

Entire text: please revisit the revised text again where formulations can be sharpened more to the point. See example for line 205.

*Thank you for your suggestion. We have made several edits to sharpen the points.*

Conclusion:
The fact that GPP is still underestimated by 20% should be reflected in the confidence description of your results. The model is missing the elevation gradient in GPP which is also reflected in the flux-tower comparison. This should be clearly stated in the conclusion.

*Thank you for your comment. We have added the following text at L 345-348 in the conclusion to summarize these points.*

*"While on average the model underestimated GPP compared to flux tower data ( ≈ 45%), we observed that the model performed well for the lower elevation sites compared to the higher elevation sites. In these simulations, variations due to the elevation gradient was not well captured as the model parameters we used were primarily developed for lower elevation sites."*

Thanks for your understanding and looking forward to see your revised manuscript.
Best wishes,
Kirsten.
* * *
*Other changes*
*We made changes to captions of Figures A2 and A3 as they appeared to be incorrect.*

*Reference*
Hurtt, G. C., Dubayah, R, Drake, J., Moorcroft, P. R., Pacala, S. W., Blair, B. J., Fearon, M. G. 2004. *Beyond potential vegetation: combining liar data and a height-structured model for carbon studies*, Ecological Applications, 14(3), 873–883
Calvo, M.M. and Prentice, I.C. 2015. *Effects of fire and CO2 on biogeography and primary production in glacial and modern climates*. New Phytol, 208: 987-994.
Moorcroft, P. R., Hurtt, G. C., and Pacala, S. W, 2001. *A method for scaling vegetation dynamics: The ecosystem demography model (ED)*, Ecological Monographs, 71(4), 557-586.
Chapin, F.S., McFarland, J., David McGuire, A., Euskirchen, E.S., Ruess, R.W., Kielland, K., 2009. *The changing global carbon cycle: Linking plant-soil carbon dynamics to global consequences*. J. Ecol. 97, 840–850.